# Characterizing quasi-biweekly variability of the Asian monsoon anticyclone using potential vorticity and large-scale geopotential height field

Arata Amemiya[1] and Kaoru Sato[2]

[1]RIKEN Center for Computational Science, Kobe, Japan
[2]Department of Earth and Planetary Science, Graduate School of Science, The University of Tokyo, Tokyo, Japan

**Correspondence:** Arata Amemiya (arata.amemiya@riken.jp)

**Abstract.** The spatial pattern of subseasonal variability of the Asian monsoon anticyclone is analyzed using long-term reanalysis data, focusing on the large-scale longitudinal movement. The air inside the anticyclone is quantified by a thickness-weighted low PV area on an isentropic surface. It is shown that the longitudinal movement of the air inside the Asian monsoon anticyclone has a timescale of one to two weeks, which is shorter than the monthly dominant timescale of the variability in the anticyclone intensity. The movement of the anticyclonic air is suggested to be largely controlled by passive advection. The typical time evolution of the variability pattern, explained by two leading EOF components of 100 hPa geopotential height, shows large-scale geopotential anomalies moving westward spanning from low to middle latitudes. This corresponds well with the rapid westward movement of low-PV air known as 'eddy shedding' and following eastward retreat of the anticyclonic air. The two EOF components can also explain the bimodal longitudinal distribution of geopotential maximum location.

## 1 Introduction

The Asian monsoon anticyclone (hereafter AMA; also known as the South Asian high or the Tibetan high) is characterized by a planetary-scale anticyclonic circulation, which persists in the upper troposphere and lower stratosphere (UTLS) region over the Eurasian continent throughout the northern summer. It is primarily driven by the upper level divergence associated with an extensive latent heat release over the southeast and south Asia induced by the monsoonal deep convection. Recently, this topic attracts increasing attention regarding its important role in the tracer transport between the troposphere and the stratosphere. The Asian summer monsoon region is considered one of the most important pathways of tropospheric tracers entering into the stratosphere. Persistent deep convection transports the air from the boundary layer to the upper troposphere, where the air is mostly confined within the AMA (Dunkerton, 1995; Dethof et al., 1999; Gettelman et al., 2004). The anomalies of various kinds of tropospheric tracers have been observed by satellite-based instruments (Randel and Park, 2006; Park et al., 2007, 2008; Randel et al., 2010; Luo et al., 2018; Santee et al., 2017) as well as in situ measurements by water vapor and ozone sondes (Bian et al., 2012) and aircraft (Gottschaldt et al., 2018). The processes responsible for the tracer transport from the AMA to the lower stratosphere have been intensively studied in recent years using chemical transport models (Park et al., 2009; Vogel et al., 2014, 2016; Pan et al., 2016; Vogel et al., 2019) and trajectory models (Chen et al., 2012; Garny and Randel,

2016). The large-scale slow upwelling over the AMA is suggested to be the dominant transport process (Garny and Randel, 2016; Vogel et al., 2019). Shorter time-scale processes associated with the subseasonal variability of the AMA, such as fast horizontal transport and turbulent mixing on an isentropic surface, also play important roles (Vogel et al., 2016; Pan et al., 2016; Gottschaldt et al., 2018; Fadnavis et al., 2018).

An isentropic potential vorticity (PV) map is a useful illustration of daily evolution of the air around the tropopause, as PV can be approximately considered a passive tracer, which is conserved in inviscid and adiabatic motion (Hoskins et al., 1985). The AMA can be identified as an area of significantly low PV surrounded by higher PV area, which correspond to tropospheric and stratospheric air, respectively. The large PV gradient near the boundary of the anticyclone is considered a mixing barrier, which keeps the tropospheric chemical characteristic of the air inside the AMA (Ploeger et al., 2015). The method of quantifying the AMA intensity as an area of PV values below a specific threshold has been used to analyze its seasonal and subseasonal variability (Randel and Park, 2006; Garny and Randel, 2013).

Aside from the quantity of total area, the horizontal structure of the AMA seen in low PV area also shows a significant variability, with frequent movement, deformation, and occasional splitting. The deformation and splitting of the AMA usually occur towards the west of the AMA center and called 'eddy shedding' (Popovic and Plumb, 2001). The possibility of spontaneous generation of such variability can be reproduced by a simple two dimensional dynamical model imposing a localized steady mass source (Hsu and Plumb, 2000). Further, Amemiya and Sato (2018) showed the characteristic longitudinally-trapped spatial structure of the variability can also be explained by a two dimensional dynamical model, with a little modification in background latitudinal thermal structure. The deformation and splitting of the anticyclone causes the horizontal stirring and irreversible mixing between the air trapped inside the anticyclone and outside stratospheric air (Pan et al., 2016; Gottschaldt et al., 2018). In addition, the shedding of low PV strip can occur eastward toward West Pacific. It also contributes to the tracer transport from the AMA to the midlatitude lower stratosphere (Vogel et al., 2016; Fadnavis et al., 2018).

Another important spatial characteristic of the variability of the AMA is the longitudinal movement of the anticyclone center. The occurrence of the longitude of geopotential maximum on 100 hPa shows a well-known bimodal distribution in a wide range of time scales (Zhang et al., 2002). The two modes are called Tibetan and Iranian modes based on the location of the maximum and usually used to classify temporal states of the anticyclone, although a recent study showed the robustness of the representation depends on the choice of reanalysis data set (Nützel et al., 2016).

Compared to the variability in the AMA intensity, which is considered to be predominantly driven by convective activity variability (Garny and Randel, 2013; Nützel et al., 2016), the mechanism of the variability involving the movement and deformation of the anticyclone has not been fully explained so far, although a few possible mechanisms have been suggested.

One mechanism is that the variability can be generated spontaneously by dynamical instability of two-dimensional flow. The strong horizontal shear of zonal wind on the northern and southern flank made the anticyclone dynamically unstable, which was originally suggested by Krishnamurti et al. (1973). The idea has been also supported by studies using nonlinear models, such as a beta-plane shallow water model with a localized steady forcing (Hsu and Plumb, 2000), and a mechanistic model in which a strong anticyclone is forced by a prescribed zonal jet over an idealized mountain (Liu et al., 2007). These studies have shown the possibility of the occurrence of westward eddy shedding without temporally varying external forcing.

Another possible mechanism is that the variability in the AMA structure including eddy shedding is forced by other localized pattern of subseasonal variability. It is suggested that the event-like westward movement of the anticyclone anomaly often preceded by the burst of deep convection in the southeast Asia (Annamalai and Slingo, 2001; Ding and Wang, 2007; Garny and Randel, 2013; Nützel et al., 2016). Ortega et al. (2017) has shown that the potential vorticity variability over the southern part of the AMA is often related to the convection variability migrating westward from the western Pacific in a quasi-biweekly timescale. Also, there is a well known teleconnection pattern through the eastward propagation of a quasi-stationary Rossby wave train along the subtropical jet (Terao, 1998; Ding and Wang, 2005, 2007; Kosaka et al., 2009; Branstator and Teng, 2017). The variability in PV structure of the AMA could be influenced by that pattern triggered in the upstream regions. However, the relative importance of these relations and intrinsic dynamics remains largely unclear.

One of the necessary step towards the understanding of the structural variability of the AMA is to objectively describe the dominant spatial pattern and its time evolution. Although many efforts have been made to extract the dominant variability patterns in Asian summer monsoon region, most analyses have focused on convection or the subtropical jet, not on the anticyclone in the UTLS as a main actor. The spatial characteristic of the variability in PV has been comprehensively described as westward eddy shedding. The concept of eddy shedding is a useful description of the significant event of the AMA variability, but it corresponds to only a part of time evolution, which actually takes place throughout the summer. It has been not clear yet if the structural variability of the AMA follows any particular pattern of time evolution or not.

Thus, the purpose of this study is to give a unified view of the dominant pattern of the variability of the AMA, incorporating the existing descriptions, namely, the event-like westward shedding of anticyclonic vortex with low PV air, and the longitudinal movement of the maximum geopotential height location, each of which has been separately discussed in different contexts so far. Based on that, this study attempts to give an implication for the responsible mechanism, which drives the variability.

The remainder of this paper is organized as follows: In section 2, the data and analysis methods used in this study are described. In section 3, the sub-seasonal variability of the AMA is analyzed by the method using low PV area, focusing on the longitudinal structure changes. In section 4, the time evolution of the dominant variability pattern of the AMA is examined using an empirical orthogonal function (EOF) decomposition of geopotential field and the relation to the pattern seen in the distribution of low PV area is discussed. Section 5 provides the summary of this paper and discussions regarding the mechanism of the variability and the relationship with other patterns in similar time scale found in previous studies.

## 2 Data and methods

### 2.1 Reanalysis and observational data

Dynamical variables from the ERA-Interim reanalysis data (Dee et al., 2011) at pressure levels with $1.5° \times 1.5°$ horizontal resolution are used for the analysis. Variables in the isentropic coordinates are obtained at every 5 K by a vertical interpolation of the original data. Analyzed time period is June-August of 1979 to 2016. Daily outgoing longwave radiation (OLR) data from the National Oceanic and Atmospheric Administration (NOAA) (Liebmann and Smith, 1996) for years from 1979 to 2016 are used as a proxy of convective activity.

## 2.2 PV-based metrics of the anticyclone

The intensity and longitudinal distribution of the AMA is quantified using a method based on isentropic PV in this study.

The method of identifying a vortex as an area enclosed by contours of a specific reference PV value has been originally developed for studies on the stratospheric polar vortex. It has been mainly used for two purposes. First, it provides the metric of the vortex intensity, which is directly related to irreversible time evolution, assuming that PV is approximately conserved (Butchart and Remsberg, 1986). Second, the edge of the polar vortex as a meridional transport boundary can be objectively detected as the maximum position of PV gradient (Nakamura, 1996; Nash et al., 1996). The calculation of the gradient is performed with respect to equivalent latitude (Norton, 1994). The advantage of these PV-based methods is that it can quantify the vortex intensity change due to diabatic and turbulent processes, regardless of the reversible perturbation caused by Rossby waves.

Similar methods have been applied to the analysis of the AMA, as it is a planetary-scale coherent vortex as large as the polar vortex. The total area enclosed by a specific PV contour is used to analyze the variability in the AMA intensity (Randel and Park, 2006; Garny and Randel, 2013). Also, Ploeger et al. (2015) attempted to objectively determine the location of transport barrier using PV and successfully showed that the barrier can be described using temporal PV value in mid-summer. The estimated location of barrier accords well with the position of discontinuity in mixing ratio of atmospheric minor species observed by satellite instruments.

However, the applicability of the methods developed for the polar vortex to the AMA is not straightforward. The AMA is not a circumpolar but a zonally-elongated elliptic vortex centered at low latitudes. Thus the theories underlying two-dimensional mixing barrier such as the effective diffusivity (Nakamura, 1996) are basically inapplicable. Moreover, the isentropic surfaces in the UTLS often have steep slopes and therefore show significantly different thickness between the troposphere and the stratosphere (Pan et al., 2012). This may cause the change in the area enclosed by a PV contour, even without external forcing processes such as deep convection.

The use of the total area inside a reference PV contour is an effective way to quantify the intensity of the anticyclone, as it measures the intensity of the vortex regardless of its location and structure. On the other hand, the variability in the location and structure of the AMA, which also occurs in a daily timescale and not measured by the total area, is also important. Thus, in this study, not only the total area of the low-PV air but also its longitudinal distribution is examined.

The total area as a function of time is described as $A_{\text{tot}}(t)$ in the following. The time evolution of $A_{\text{tot}}(t)$ is derived from the PV tendency equation (Butchart and Remsberg, 1986; Garny and Randel, 2013) as follows,

$$\frac{d}{dt}A_{\text{tot}}(t)_{q \leq q_0} = \oint_{q=q_0} \left( -q \frac{\partial \dot{\theta}}{\partial \theta} + \dot{\theta} \frac{\partial q}{\partial \theta} \right) \frac{dS}{|\nabla_\theta q|} + \int_{q \leq q_0} \boldsymbol{\nabla} \cdot \hat{\boldsymbol{u}} \, dA + (\text{subgrid scale mixing term}) \tag{1}$$

where $\dot{\theta}$ is the potential temperature tendency by adiabatic processes, $\boldsymbol{u}$ is horizontal wind vector, $dA$ and $dS$ are respectively an area element and a line element of contours surrounding $A(t)$. Variable $\boldsymbol{u}$ is decomposed into resolved (ˆ) and unresolved (′) components as $\boldsymbol{u} = \hat{\boldsymbol{u}} + \boldsymbol{u}'$. The terms on the right hand side are called the generation term, divergence term, and mixing term, respectively (Garny and Randel, 2013). Although the direct quantification of each term from gridded reanalysis data is

difficult, it was shown by Garny and Randel (2013) using a free-running general circulation model that the first and second terms are well correlated and hence the second term alone can be used as a proxy of convective activity.

To quantify the longitudinal movement of the low PV air, a function of longitude and time $L(\lambda, t)$, which has a unit of length, is defined so that its longitudinal integration gives a low-PV area.

$$A(t)_{q \leq q_0} \quad = \quad \int_{\lambda_w}^{\lambda_e} L(\lambda, t) d\lambda, \tag{2}$$

where $\lambda_w$ and $\lambda_e$ are arbitrary longitudes. The partial area of low PV air of the AMA on the west of a specific longitude $\lambda_0$ is obtained by the integration from the western boundary of the calculation domain to $\lambda_0$ denoted as $A_{\text{west}}(t)$. The integration over the entire domain produces $A_{\text{tot}}(t)$.

Another treatment that is newly introduced in this study is the weighting of low-PV area by equivalent thickness in isentropic coordinates $\sigma = -g^{-1} \partial p / \partial \theta$. Large variation of $\sigma$ in time and latitude significantly modifies the budget calculation both in seasonal mean and in subseasonal variability, as was indicated by Pan et al. (2012). In this study, instead of using $L$ and $A$, thickness-weighted quantities denoted by $\hat{L}$ and $\hat{A}$ are used. The weighted low-PV area $\hat{A}$ is defined as follows,

$$\hat{A}(t)_{q \leq q_0} \quad \equiv \quad \int_{q \leq q_0} \sigma dA = \int \int \sigma r^2 \cos\phi d\lambda d\phi, \tag{3}$$

where r is the radius of the Earth. $\hat{A}$ has an unit of mass divided by a temperature unit. Thus this quantity can be interpreted as a total mass of the air inside the AMA on an isentropic layer. $\hat{L}$ is related to $\hat{A}$ in a similar way as follows,

$$\hat{A}(t)_{q \leq q_0} \quad = \quad \int_{\lambda_w}^{\lambda_e} \hat{L}(\lambda, t) d\lambda \tag{4}$$

The equation for the tendency of mass-weighted total area $\hat{A}_{tot}(t)$ takes the following simpler form. See Appedix B for the derivation.

$$\frac{d}{dt} \hat{A}_{\text{tot}}(t) \quad = \quad \oint_{q=q_0} \left( -q \frac{\partial \dot{\theta}}{\partial \theta} + \dot{\theta} \frac{\partial q}{\partial \theta} \right) \frac{\sigma dS}{|\nabla_\theta q|} - \int_{q \leq q_0} \frac{\partial}{\partial \theta}(\sigma \dot{\theta}) dA + (\text{unresolved term}) \tag{5}$$

Further, the budget of the mass-weighted partial area to the west of specific longitude $\lambda_0$ changes due to mass-weighted zonal flux of low-PV air $\hat{F}(\lambda_0)$, in addition to the nonconservative terms :

$$\frac{d}{dt} \hat{A}_{\text{west}}(t) \quad = \quad -\hat{F}(\lambda_0) + \oint_{q=q_0, \lambda \leq \lambda_0} \left( -q \frac{\partial \dot{\theta}}{\partial \theta} + \dot{\theta} \frac{\partial q}{\partial \theta} \right) \sigma dS - \int_{q \leq q_0, \lambda \leq \lambda_0} \frac{\partial}{\partial \theta}(\sigma \dot{\theta}) + (\text{unresolved term}) \tag{6}$$

$$\hat{F}(\lambda) \quad = \quad \int_{q \leq q_0} u \sigma r d\phi \tag{7}$$

The flux $\hat{F}(\lambda_0)$ represents the mass flux of low PV air in the zonal direction at the longitude $\lambda_0$, integrated over the latitude where the PV is below the threshold. Comparing the left hand side and the flux term on the right hand side, the relative contribution of the movement of anticyclonic air due to conservative processes can be quantified.

The Eq. 5 and Eq. 6 have no horizontal divergence term, which is present on the right hand side of Eq. 1. This implies that, by the thickness weighting, $\hat{A}$ changes only when there is non-conservative forcing such as diabatic heating. Therefore, the $\hat{A}$ can be interpreted as an equivalent 2-d representation of the air parcel. In other words, this formulation separates the effect of horizontal divergence induced by the diabatic heating from that induced by compression or tilting of the air parcel by the steep slopes of isentropic surfaces. The tendency of $\hat{A}_{\text{west}}$ is mostly explained by the longitudinal flux term $\hat{F}(\lambda)$ when other nonconservative processes over the area of $\hat{A}$ are negligible. Note that this formulation, based on the thickness-weighted area of low PV air, is different from that based on the thickness-weighted PV budget and PV flux introduced by Ortega et al. (2018). The flux term $\hat{F}(\lambda)$ represents the mass flux of low PV air, not the PV flux. This means, given that the threshold PV value properly reflects the mixing barrier, the flux of low PV air implies how atmospheric chemical tracers are distributed, even when a local value of PV changes in the regions where PV is below the threshold. As an illustrative example, Fig. 1a shows a PV field on the 370 K isentropic level on a specific day. The area of PV values below 2 PVU is hatched by red dots. Figure 1b shows the distribution of horizontal winds and thickness $\sigma$ inside the AMA. Strong anticyclonic circulation is seen along the boundary of the area. Larger values of thickness are found in the northern part of the area compared to the southern part, indicating the effectiveness of thickness-weighting.

The analysis is generally sensitive to the choice of reference isentropic level, PV threshold, and calculation domain. In this study, the horizontal area 10°W–160°E, 10°N–50°N is examined. The reference PV value, isentropic surface, and the analysis domain in this study are compared to the previous studies, which used the similar method (Randel and Park, 2006; Garny and Randel, 2013; Ploeger et al., 2015) in Table 1. Budget calculations are performed for three respective isentropic levels of 360 K, 370 K and 380 K using the PV thresholds shown in Table 1. A detailed analysis is mainly performed for July and August on the 370 K level, where the PV gradient is the largest. The detailed reasoning for these reference values is described in Appendix A.

## 3   The variability of the AMA seen as low PV area

### 3.1   Total thickness-weighted area

Figure 2 shows the time series of thickness-weighted low PV area defined on 360 K and 370 K isentropic levels, along with the time series of area-averaged OLR over 15°N – 30°N, 60°–120°E from June to August. The result for a specific year 2016 is shown as an example. Thick lines represent lowpass-filtered 1979–2016 mean with a cut-off length of 31 days. The mean seasonal evolution of weighted low PV area during summer has a peak in the middle of July. The total low PV areas measured at 360 K and 370 K in 2016 both fluctuate with a time scale near 30 days including minima in late June and late July, and maxima in mid July and mid August. There is also a steep maximum in the middle of June both in low PV area and OLR, implying an event with a shorter timescale. There is a clear correspondence between fluctuations in total low PV area and area-averaged

OLR in this year with about a few days lag, especially for 360 K, as pointed out by previous studies (Randel and Park, 2006; Garny and Randel, 2013).

## 3.2 Longitudinal movement of the air inside the AMA

Previous studies used the total area $\hat{A}_{\mathrm{tot}}$ has been used, without weighting, to analyze the AMA variability and found the dominant timescale around 30 days. However, the use of total area does not take account of the zonal displacement and deformation of the AMA which may have different timescales and patterns, such as those described as the 'eddy shedding'. The longitudinal distribution of low PV area is analyzed in our study as in the following.

Figure 3 shows an average of the zonal flux of thickness-weighted area of low PV air (Eq. 7) as well as the standard deviation as a function of the longitude on each of the 380 K, 370 K, and 360 K level for July and August of 1979-2016. At 370 and 380 K, the mean zonal flux is slightly eastward. The standard deviation is much larger and maximized in the longitudinal region of about 40°E – 100 °E. This implies that the area of low PV air oscillates zonally within this longitudes while time-mean zonal movement of air is relatively insignificant on this level. Note that the mean flux of low PV air does not need to be zero because it can be balanced with nonzero PV source/sink by differential heating which is significant in seasonal mean and largely depends on the vertical levels. The important feature observed both in Figs. 3a and b is large fluctuations in the zonal flux of low PV air as shown by the large standard deviations. This implies that a large part of the airmass inside the AMA can move eastward and westward. This is consistent with the results of trajectory analysis by Garny and Randel (2016), in which passive tracers released within the anticyclone tend to be trapped inside for about a month on average. At 360 K, the mean zonal flux is largely negative between 30°E and 120°E. The positive zonal flux divergence in the eastern part is likely generated by convective forcing, while the positive flux convergence in the western part would be balanced by the sink in low PV area due to large vertical gradient in radiative heating/cooling around the 360 K level.

Figure 4 shows the longitude-time cross section of thickness-weighted anticyclone area $\hat{A}$ calculated for each longitudinal grid interval and its zonal flux $\hat{F}(\lambda, t)$ at 360, 370, and 380 K. Results for the summer months of 2016 are shown for example. Frequent zonal movement of the anticyclone air with a sub-monthly timescale is clearly seen. Similar longitude-time plots are found in previous studies, but most of them showed only variables averaged over a fixed range of latitude in the southern part of the AMA. Thus they may miss the longitudinal structure change after eddy shedding. The view based on the low-PV area, first introduced by Garny and Randel (2013), is useful to capture the zonal movements of the AMA regardless of its latitudinal position. Figure 4 is similar to thier Fig. 6, except that the weighted values are used and the zonal flux $\hat{F}$ is added. At 360 K, the longitudinal flux is mainly westward. At 370 K and 380 K, in contrast, the pulses of both westward and eastward flux occur alternately in July in the region of 30 °E–120 °E. The budget of the low PV air shows different characteristics at each of these levels. The alternate eastward and westward movements at 370 K imply that the variability is more like oscillatory behavior rather than dissipative westward eddy shedding as reproduced by a two dimensional model (Hsu and Plumb, 2000).

The daily evolution of the time tendency of partial thickness-weighted area $\frac{d}{dt}\hat{A}_{west}$ at 360 K, 370 K and 380 K on the west of 60 °E and the contribution from the zonal flux $-\hat{F}$ through that longitude are shown in Fig. 5. At 370 K, the two curves correspond with each other very well, except in the beginning of June when the noise in PV-based definition around

the southern boundary could be large. The correlation coefficient between these two terms calculated from the time series for 1979-2016 is as high as 0.663. This result means that the oscillation of the AMA at this level is mostly due to a simple zonal advection, and other effects such as dissipation by turbulent mixing are secondary. At 360 K, the two curves still show the variability synchronized with each other, but with a large offset. This implies the existence of a large sink of low PV area at 360 K. At 380 K, the two curves also accord well with each other, although large disagreements are occasionally observed. This may imply sporadic nonconservative processes at this level. Note that the disagreement may be just due to the error in calculating $\hat{A}_{west}$, which is more likely to occur at 380 K than at lower levels (See Appendix A). While the whole picture to explain the budget of low PV area at these levels is more complicated than what is seen at a single level, we consider the longitudinal oscillatory behavior with a submonthly time scale is one of the important features of the variability of AMA, and the zonal flux $\hat{F}_{\lambda=60^\circ}$ at 370 K as a representative variable for it. This variable will be used as a proxy of the zonal movement of the AMA in the composite analysis in next section.

Figure 6 shows the power spectrum at 370K of the zonal flux $\hat{F}_{\lambda=60^\circ}$ as a 38-year mean. There is a broad peak between about 9 and 20 days, and no peak is found around 30 days, which corresponds to the dominant period of the variability of the AMA intensity shown in Garny and Randel (2013). This supports the idea that the variability pattern with the zonal movement of the AMA is effectively separated by extracting short-period components of the variability, including the quasi-biweekly timescale mentioned in previous studies. For this reason, the quasi-biweekly time scale is focused on in the following analyses.

## 4   The life cycle of quasi-biweekly oscillation of the anticyclone

### 4.1   EOF decomposition

The life cycle of the dominant large scale pattern of the subseasonal variability of the AMA is examined using the empirical orthogonal function (EOF) decomposition. The EOF analysis is applied to the daily-mean geopotential anomaly in the domain covering the AMA using ERA-Interim reanalysis data. Before calculating EOFs, anomalies are filtered with a band-pass filter within time periods of 5–20 days, normalized at each grid point, divided by their standard deviations and weighted by the square root of grid areas. In other words, decomposition is performed for correlation matrix weighted by each grid area instead of covariance matrix. The normalization is effective to reduce the effect of the latitudinal dependence of geopotential height perturbation amplitude. Otherwise the perturbation patterns would be concentrated at midlatitudes and coherent low latitude features may be missed. The analysis is made for the regions of 0°E–150°E, 10°N–50°N at 100 hPa and July and August of 1979–2016. EOFs for longer period such as June to September, for slightly different levels such as 150 and 200 hPa, or for a slightly different horizontal domain do not differ much. It was confirmed that extended EOF and complex EOF analysis provides essentially similar spatial patterns (not shown). Thus, only the results obtained by a standard EOF analysis are shown.

The first two EOF components are dominant and sufficiently separated from others by North's rule of thumb (North et al., 1982). The partial variance explained by those two components are about 15% and 13%, respectively. The spatial structures of geopotential height of the two components are shown in Fig. 7. Both EOF1 and EOF2 have large-scale longitudinal wavy patterns approximately from 0°E to 120°E over low and mid latitudes. The combination of these two components explain about

30 % of zonally averaged total perturbation variance. Note that this latitudinal structure having large amplitudes both in low and mid latitudes is different from well-known patterns extracted by EOFs for meridional wind perturbations (Kosaka et al., 2009) or the composite analysis for the extreme events of positive geopotential height anomaly averaged over 35-45 °N, 55-75 °E (Ding and Wang, 2007). The pattern extracted in this study shown in Fig. 7 has larger scales than those previous studies. This is likely due to the choice of geopotential height anomaly for the EOF analysis which tends to favor larger-scale structure than other variables such as meridional wind.

The time series of principle components (PCs) 1 and 2, corresponding to two leading EOFs, have significant lag correlation with each other, as shown in Fig. 8. PC1 is lagged behind PC2 by about 3 days. As EOF2 has the structure which is out of phase by about a quarter cycle with EOF1, indicating the dominance of westward propagation of wavy patterns.

Eight phases are defined based on normalized PC1 and 2 time series, as illustrated in Fig. 9. The daily states are labeled as each of these phases when geopotential perturbation amplitude projected onto that two dimensional phase space exceeds unity. The summary of phase progress statistics is shown in Fig. 10. For more than 50% cases, the phase progresses to the next within one day. And for more than 60% cases it does within two days. It is also shown that the reverse progress rarely occurs. This implies the transition from a phase to the next one mostly takes place within two days, which is consistent with lead time in Fig. 8 and corresponds to the quasi-biweekly dominant timescale found in section 3. Based on these facts, the characteristics of the time evolution of the AMA variability pattern are examined by composite mean maps for each phase.

## 4.2 Composite life cycle of the AMA variability pattern

First, the life cycle of the extracted disturbance is examined in terms of the intensity, location and structure of the AMA based on the low PV area with a PV threshold of 2 PVU on the 370 K isentropic level (see section 2.4). Figure 11 shows maps of the frequency of existence of the low PV air for each phase. The area with a percentage greater than 60 % are color-shaded. As the phase progresses from phase 6 through phase 1, a large portion of high frequency area moves westward and reaches around 30°E. After that, the high frequency area drifts slightly northward and moves eastward back from phase 2 to phase 4. This zonal oscillatory behavior is almost consistent with the large fluctuation of zonal flux of low PV air observed in longitudes from 30°E to 110°E in Fig. 3. The zonal movement with the phase progress is quantitatively shown as a composite mean of the thickness-weighted zonal flux of low PV air at 370 K at 60°E (Fig. 12). The flux has negative (i.e. westward) peak around phase 6 and positive (eastward) peak around phase 3. Fig. 13 shows the mean area of the whole AMA and partial area at 370 K to the west of 60°E for each phase. Standard deviation is also shown by dashed curves. The western part fluctuates with phase, as it is largely controlled by the zonal flux in weekly timescale (Fig. 5). In contrast, the total area does not show significant dependence on the phase. As the total area is likely to be controlled by the total intensity of the thermal forcing, this implies that the variability in the thermal forcing intensity is not a key factor for this variability pattern. This supports the idea that the variability in the biweekly timescale is determined by internal dynamics.

Next, the spatial characteristics of geopotential and the variables related to convective activities during the phase progress are examined. Composites of geopotential (black contours) and its anomaly from climatology (color shades) on 100 hPa along with OLR anomalies [red(positive) and green(negative) contours] are shown in Fig. 14. As already seen in Fig. 7,

geopotential anomaly exhibits large-scale wave-like pattern propagating westward along the subtropical jet about 40°N – 50°N. The geopotential anomalies are extended southeastward to low latitudes. The positive geopotential anomaly along 30°N is located at about 70°E at phase 5, 50°E at phase 6, 40°E at phase 7, and 30°E at phase 8. The westward movement of positive geopotential anomaly follows roughly the movement of high occurrence of low PV area from phase 5 to phase 8 (Fig. 11).

Statistically significant OLR anomalies are found mainly over Tibetan Plateau, southern China, and the region from northern Bay of Bengal to northern India. The anomalies over the Tibetan Plateau is in phase with geopotential anomalies of the same sign. Over other two regions, the relation of the OLR anomalies with geopotential anomalies and with PV distribution (Fig. 11) is less clear. The observed convection variability may occur in response to the large scale dynamical variability, and/or have an influence on the dynamics. This point will be discussed in the next section.

Fig. 15a and 15b show the geopotential anomalies averaged over 15°N–25°N and 35°N–45°N, respectively, in the longitude-pressure cross sections. Color shading shows the geopotential anomalies from the climatology, whereas contours show the anomalies from longitudinal mean to indicate the AMA center location. Only results for phases 1 to 4 are shown, as the features for the rest of the phases appear close to the negative counterparts. The longitude-pressure structure of the anomalies is almost barotropic in both the low and middle latitudes. The amplitude of the anomalies is maximized around 100 hPa for the low latitudes and around 150–200 hPa for the midlatitudes, reflecting the difference in the tropopause height. The level of the maximum roughly corresponds to the tropopause level. Thus it is considered that the pattern is essentially trapped by the tropopause, as is the AMA itself (Popovic and Plumb, 2001). For the low latitudes, significant anomalies are observed down to about 400 hPa, close to the level where the convective thermal forcing is maximized. In contrast, vertical structure is the deeper at midlatitudes.

## 4.3    The relation with the bimodality in anticyclone center location

The bimodality of the center longitude seen at 100 hPa is one of the important characteristics of spatial variability of the AMA. The relation between the bimodality and the quasi-biweekly AMA variability pattern defined in this study is examined.

Using the ERA-Interim reanalysis data from July to August of 1979-2016, the longitudinal distribution of the AMA center, that is defined as the location of daily mean geopotential height maximum, is calculated. The total distribution is shown in the bars in Fig. 16. The bimodal distribution with peaks around 60 °E and 90 °E is observed. The distribution is also calculated separately for each phase, as shown in curves in different colors in Fig. 16. Numerals show the location of peak longitudes for respective phases. There is a clear phase dependence of the distribution. Phases 1 to 3 favor the eastern location around 90°E of the AMA center, while phases 5 to 8 favor the western location around 60°E. This phase dependence can be mostly explained by the spatial structure of EOF2 shown in Fig. 7(b). The partial variance explained by EOF2 component is sufficiently large to enable large positive and negative EOF2 components to contribute to eastward and westward displacement of the AMA center, respectively.

## 5    Summary and Discussion

In this study, the subseasonal variability of the AMA, which includes longitudinal movements on a quasi-biweekly time scale
was examined. The analyses were performed from two different perspectives, that is, the movement of the air inside the AMA
defined as thickness-weighted low PV area and an EOF decomposition of normalized geopotential anomaly field on 100 hPa.

The zonal distribution of thickness-weighted low PV area and its zonal flux was calculated using ERA-Interim reanalysis data
from 1979 to 2016. The longitudinal distribution of low PV area in mid-summer exhibits a significant temporal fluctuation.
The budget analysis revealed that the tendency of the partial thickness-weighted low PV area on the west of the specified
longitude of 60 °E is mostly controlled by the zonal flux entering the domain on a subseasonal time scale. This suggests
that the variability is mostly controlled by the large scale dynamics, and other nonconservative processes such as turbulent
dissipation and diabatic heating by radiation and/or convection have secondary roles on this time scale. Thus the variability can
be characterized as event-like pulses of zonal flux of low PV air at the longitude around 60 °E.

The large-scale variability pattern in geopotential height on a pressure level around the tropopause was reconstructed by
the first two EOF components. They explained about 30% of the total variance and are significantly separated from other
components. They had large scale anomaly patterns spanning from middle to low latitudes with comparable amplitudes and
a significant lead-lag correlation with each other. The reconstructed pattern by these two components showed a westward-
moving large scale geopotential anomaly. By defining the phase of this pattern based on a two-dimensional phase space by
these leading modes, spatial structures of variables at each phase in the whole life cycle were examined. The distribution of the
occurrence frequency of low PV on 370 K level indicated clear zonal oscillation of the air inside the AMA as phase progresses.
There was a significant relationship between the phase and zonal flux of low PV air, as the rapid westward movement around
60 °E at phase 6 in Fig. 11 corresponds to large westward flux of low PV air in Fig. 12. Composite of geopotential height shows
the westward-propagating large-scale anomaly pattern along the subtropical jet. The vertical structure is nearly barotropic in
both middle and low latitudes. The influences down to the level of middle troposphere was observed at the midlatitude, whereas
at the low latitude disturbances are largely trapped above 300 hPa.

The variability pattern revealed in this study has a robust tendency of cyclic time evolution (Fig. 10) in which a quasi-
biweekly time scale is dominant. Thus an important question is what drives this variability and determines the time scale.
The driver of the AMA variability has been discussed in previous studies focusing mainly on seasonal evolution and longer
period variability patterns. For the variability in the anticyclone intensity with a monthly time scale, the essential role of the
variability in convection in south to southeast Asia has been suggested (Garny and Randel, 2013; Nützel et al., 2016). However,
it is not straightforward the similar relationship applies to the quasi-biweekly variability focused on in this study. The causal
relationship between convective variability and the UTLS circulation variability in the quasi-biweekly timescale remains an
important question. There have been a few studies, which suggest variability patterns coupled with convection anomalies of
smaller spatial scale over the area including the Tibetan Plateau, east Asia, south Asia, and the western Pacific. For example,
Fujinami and Yasunari (2004) suggested a cyclic pattern of convective anomalies propagating clockwise from Tibetan Plateau
via south China and northeast India, along with Rossby wave train over the subtropical jet. Also, a recent study by Ortega et al.

(2017) has examined the coupling between quasi-biweekly variabilities in tropospheric convection and the UTLS dynamics, suggesting the possibility that the latter one leads the former. However, as they used area-averaged PV over southern India for a metric of upper tropospheric disturbances, their results have captured the pattern significant for the southeastern part of the area of AMA, not the whole extent of AMA, which includes midlatitudes and west Asia. In this study, the statistically significant pattern of OLR anomalies was found in the composite analysis based on the EOF leading components of dynamical field perturbations (Fig. 14). This anomaly pattern of convection may be the response to the dynamical variability, or rather the driver of the variability. A recent study by Wei et al. (2019) suggested the latter possibility based on their composite analysis focusing on the longitudinal movement of the AMA center. However the spatial location of the OLR anomalies corresponding to each of the phase of the AMA variability found in this study does not necessarily support it. For example, in phase 6, there are significant negative OLR anomalies over southern China and northeastern to northern India. This feature indicates the intensified convection and the explanation of low PV area by upper tropospheric horizontal divergence over these areas. However, at this phase, the maximum low PV area probability in Fig. 11 is located around 50-80 °E, to the west of the area of intensified convection. This mismatch implies that the westward movement of low PV area is not forced or triggered by a temporal burst of monsoonal convection over the northeastern to northern India, provided that this phase progress based on the EOF analysis properly captures the characteristic time evolution of dynamical fields associated with the variability pattern. The relationship between the anomaly pattern found in this study and that of previous studies should be explored in future studies.

The subseasonal variability pattern can be driven by dynamical instability of two dimensional anticyclonic flow, as mentioned in section 1. The essentially barotropic structure of the anomalies seen in Fig. 15 supports the validity of a conceptual two dimensional model to explain the dynamics. The analysis in this study using thickness-weighted low PV area and its zonal flux showed the pulse of westward movement of low PV air characterizing the variability. This behavior corresponds to spontaneous eddy shedding reproduced by the two dimensional model in a previous study (Hsu and Plumb, 2000). However, there is an essential difference between their modeled eddy shedding and the observed variability in terms of the budget of low PV area. Whereas anticyclonic eddies dissipate after westward shedding in the conventional dynamical model, the low PV area in reality does not dissipate but mostly returns eastward and forms the oscillatory pattern seen in Fig. 11 and Fig. 12. A recent study attempts to explain this behavior, using a modified two-dimensional dynamical model, which includes the effect of latitudinally-varying tropopause structure (Amemiya and Sato, 2018).

The zonal oscillation of the AMA viewed as low PV area in this study is directly linked to the oscillation of the mixing ratio anomalies of various atmospheric minor constituents, as PV approximately acts as conserved quantity as well. Irreversible tracer transport through the AMA occurs with several dynamical or physical processes. Those are dependent on the temporal structure or position of the AMA during the subseasonal variability. For example, the large scale upwelling, which transports the air into the tropical lower stratosphere, may correspond to the temporal position of the AMA. Turbulent mixing can be enhanced in the process of eddy shedding. Also, occasional westward or eastward shedding of tropospheric air out of the AMA, which contributes to the transport to the midlatitude stratosphere (Vogel et al., 2016), may be dependent on the phase of large scale variability described in this study.

Additionally, the understanding of the dynamics of the AMA and its relation to tropospheric weather patterns is practically important. The subseasonal variability of the Asian summer monsoon is one of the most essential factors in predicting the risk of high impact weather such as heavy rainfall and droughts in south and southeast Asia. For the practical purpose, describing the dominant variability patterns and their typical time evolution provides a useful framework for a subseasonal prediction, as has been successfully applied to Madden-Julian Oscillation. Recent studies have found several dynamical predictors for heavy rain events (Ding and Wang, 2009) and introduced real-time multivariative indices for the variability of the Asian summer monsoon (Lee et al., 2013). The pattern discussed in this study based on low PV area and geopotential anomalies focuses on a wider area from middle east to east Asia including the Tibetan Plateau. The relative importance of this variability pattern for local precipitation prediction and the relation to existing patterns is an interesting topic for future study.

*Data availability.* The ERA-Interim data was downloaded from the ECMWF data server (http://apps.ecmwf.int/datasets/, last access: 26 April 2020). Daily outgoing longwave radiation (OLR) data was downloaded from the NOAA PSL server (https://psl.noaa.gov/data/gridded/data.interp_OLR.html, last access: 26 April 2020).

**Appendix A: The choice of reference PV, isentropic level and southern boundary latitude of the domain**

In this study, the area of the anticyclone is defined as the area in which PV is lower than a specific reference value. The domain used for this analysis should be specified by certain ranges of longitude and latitude where the AMA is typically found. The proper choice of such longitude and latitude ranges, as well as the reference PV value and isentropic level, is not a trivial issue, because the boundary of the AMA is not always well-defined by a fixed value of PV, especially on its southern flank. As seen in Fig. 1, PV values near the equator are as low as that inside the AMA, although the equatorial UTLS air generally have different origin and does not mix with the air inside the AMA easily, as they are usually separated by the air in low latitudes with larger PV values. Ideally, by choosing sufficiently low reference PV value and proper position of the southern boundary of the domain, the AMA can be detected as an isolated area of low PV within the domain. However, due to the large subseasonal variability and seasonal evolution of the AMA both in PV value and structure, it is sometimes difficult to isolate the area of the AMA from the equatorial air. When the southern boundary of the domain is located too close to the equator, the equatorial air, which is supposed to be stratospheric origin may be counted as the area inside the AMA. In contrast, when it is located at higher than the optimal latitude, part of southern portion of the AMA will be excluded. In such a case the AMA intensity is underestimated and the important response to the low latitude convection might be missed. Additionally if the reference PV value is too large, it is not always possible to separate the AMA and the equatorial air by the closed PV contour. Therefore, although these sources of error can not be completely excluded, it is worthwhile to show how and to what extent they can be minimized by the optimal choices of the domain boundary and reference PV and isentropic surface values.

In the following, the largest source of error is considered to be generated from the choice of the southern boundary latitude of the domain and a reference PV value, while other boundary positions are fixed. The northern boundary is fixed at 50°N. The

eastern and western boundary are set to 160°E and 10°W, respectively, following Ploeger et al. (2015). It is confirmed that the sensitivities of calculated AMA areas to these values are not significant.

The optimal southern boundary latitude and the reference PV value are explored as follows. On each of the isentropic surfaces
360, 370, and 380 K, the frequency of occurrence of grid points with PV lower than the reference within the longitudinal range (10°W to 160°E) is calculated as a function of latitude. Calculation is performed for each month from June to September, using the 6 hourly ERA-Interim reanalysis data from 1979 to 2016.

Figure A1 shows the results on each of three isentropic surfaces for each month. Generally, the occurrence of low PV air has a peak in subtropical latitudes separated from the high occurrences near the equator. The relative error of the AMA area
calculation is implied by the significance of isolation of such a peak. Sufficiently high occurrence of PV lower than the reference value around the subtropical latitudes and infrequent occurrence on the equatorial periphery of the AMA are desirable. Such a contrast is the most clear in July and August at 370 K, when the value around 2 PVU is used as the reference. In June, the AMA areas are almost similar to those in July, although there are higher occurrences in low latitudes. The latitude of maximum occurrence, which roughly corresponds to the AMA center is located southward compared to that in July, and the
separation from the equatorial air is less clear. In September, the contrast in occurrence between low latitudes and mid latitudes becomes obscure for most reference PV values. Such behavior with respect to seasonal transition is consistent with the seasonal evolution of the AMA shown in monthly climatology of PV on 370 K isentropes in Fig. A1. The similar seasonal evolution can be observed at other isentropic levels. At 380 K, less significant maximum occurrences in mid latitudes are found. At 360 K, the whole pattern is shifted to lower latitudes and the minimum in low latitudes is more obscure. Considering these results, the
value around 2.0 PVU on 370 K isentropic level is implied to be the best reference value. On 360 K and 380 K, the respective reference values should be around 0.5 PVU and 3.5 PVU to decribe the anticyclonic vortex while minimizing the noise by the southern boundary.

Those values are also compared with the barrier PV values on each isentropic level, determined objectively following the method by Ploeger et al. (2015). Their criterion to detect mixing barrier is based on the maximum of PV gradient with respect
to equivalent latitude defined base on PV (see their section. 4 for the detailed methodology). Using the long-term reanalysis data from 1979 to 2016, we calculated the barrier PV value for each day on each of 360, 370 and 380 K isentropic levels.

Figure A2 summarizes the result. The median (50 percentile) and 15 and 85 percentiles of barrier PV values, along with the ratio of barrier detection in number (%), are shown for each 10 or 11-day month part of the boreal summer season. We found that 370 K is the most favorable isentropic level to detect barrier objectively, with percentages above 60% throughout July and
440 August, although Ploeger et al. (2015) have shown that 380 K provides the most significant barrier feature. This discrepancy can be explained considering that the detectability can vary in interannual and long period intraseasonal timescale, reflecting large dynamical variability. On 360 K, the percentage of barrier detection has lower values compared to 370 and 380 K but still larger than 40 % in July. In July and August, the range of barrier PV values are almost unchanged with respect to time, while in June lower values are more dominant. Median PV values on each of 360, 370 and 380 K levels are near 0.5, 2.0 and
445 3.5 PVU, respectively. These values are close to optimal values implied by Fig. A1 to detect anticyclonic air sufficiently wide and also separated from the equatorial air.

Supported by these results, we chose the calculation domain and reference values in the following. The domain is chosen to be inside 12–50°N, 10°W–160°E. The reference PV value used for analyses were chosen to be 2.0 PVU on 370 K. The high percentage of barrier detection around this value implies the relevance of the underlying concept of isolated strong nearly-inviscid anticyclone, in which low PV air is surrounded by distinct boundary with steep PV gradient. For the purpose of the comparison of zonal anticyclone area flux on different levels, reference values 0.5 and 3.5 PVU were also used respectively on 360 and 380 K. Most analyses were performed for July and August, when the anticyclone is the most intense and the PV-based definition is the most relevent. Three-months data from June to August is only used for the spectrum analyses.

## Appendix B:  Derivation of the tendency equation for the thickness-weighted low PV area

Equation (3) can be derived in a way similar to the derivation of Eq. (13) of Butchart and Remsberg (1986). For consistency, their notation is used in the following. They begin with the small change in the area $\Delta A(t)_{\chi \geq \chi_0}$ enclosed by an isopleth $\chi = \chi_0$, denoted by $\Gamma$, with respect to the change in the contour position $\Delta \boldsymbol{x}$. Note that they assumed that the value of $\chi$ increases inward of the area.

$$-\oint_{\Gamma} \Delta \boldsymbol{x} \cdot \frac{\nabla_\theta \chi}{|\nabla_\theta \chi|} ds = \Delta A(t)_{\chi \geq \chi_0} \tag{B1}$$

To extend this formulation to the thickness-weighted area $\hat{A}(t)_{\chi \geq \chi_0}$, there needs an additional term for thickness change $\Delta \sigma$ in the left hand side,

$$\int_{\chi \geq \chi_0} \Delta \sigma dA - \oint_{\Gamma} \Delta \boldsymbol{x} \cdot \frac{\nabla_\theta \chi}{|\nabla_\theta \chi|} \sigma ds = \Delta A(t)_{\chi \geq \chi_0} \tag{B2}$$

Using the equation for thickness

$$\frac{\partial \sigma}{\partial t} + \nabla_\theta \cdot (\sigma \boldsymbol{v}) = -\frac{\partial}{\partial \theta}(\sigma \dot{\theta}), \tag{B3}$$

the first term on the left hand side of Eq. (B2) can be rewritten as follows.

$$\int_{\chi \geq \chi_0} \Delta \sigma dA = \Delta t \left[ -\int_{\chi \geq \chi_0} \nabla_\theta \cdot (\sigma \boldsymbol{v}) dA - \int_{\chi \geq \chi_0} \frac{\partial}{\partial \theta}(\sigma \dot{\theta}) dA \right] \tag{B4}$$

$$= \Delta t \left[ \oint_{\Gamma} \boldsymbol{v} \cdot \frac{\nabla_\theta \chi}{|\nabla_\theta \chi|} \sigma ds - \int_{\chi \geq \chi_0} \frac{\partial}{\partial \theta}(\sigma \dot{\theta}) dA \right] \tag{B5}$$

where $\boldsymbol{v}$ is two-dimensional velocity, and $\nabla_\theta$ is two-dimensional gradient on an isentropic surface.

The second term on the right hand side is transformed using $\Delta \boldsymbol{x} \cdot \nabla_\theta \chi \simeq -\Delta t \cdot \partial \chi / \partial t$ as shown in Butchart and Remsberg (1986). Then by applying $\Delta t \to dt$ we obtain the following,

$$\frac{d}{dt} \hat{A}(t)_{\chi \geq \chi_0} = \oint_{\Gamma} \left( \frac{\partial \chi}{\partial t} + \boldsymbol{v} \cdot \nabla_\theta \chi \right) \frac{\sigma ds}{|\nabla_\theta \chi|} - \int_{\chi \geq \chi_0} \frac{\partial}{\partial \theta}(\sigma \dot{\theta}) dA \tag{B6}$$

As the first integral contains the advection term, the bracket can be replaced with a nonconservation term $F$.

$$\frac{\partial \chi}{\partial t} + \boldsymbol{v} \cdot \nabla_\theta \chi = F \tag{B7}$$

When $\chi$ is potential vorticity and the area $A$ is defined to have potential vorticity below the reference value, Eq. (3) is ob-
475 tained. The subgrid scale mixing term in Butchart and Remsberg (1986) comes from the difference between the true divergence term and the divergence term calculated from resolved variables. Although our equation does not have divergence term, we consider it is still better to include unresolved effect such as subgrid scale mixing, which is included in $F$. Then, when $\chi$ is potential vorticity and the area $A$ is defined to have potential vorticity below the reference value, using

$$F \;\; = \;\; -q\frac{\partial \dot{\theta}}{\partial \theta} + \dot{\theta}\frac{\partial q}{\partial \theta} + (\text{unresolved term}) \tag{B8}$$

thus we obtain Eq. (3).

Equation (3) can also be derived from the general mass conservation expression in a PV-$\theta$ coordinate introduced in Nakamura (1995);

$$\left(\frac{\partial m}{\partial t}\right)_{q,\theta} + \left(\frac{\partial \mathcal{M}(\dot{q})}{\partial q}\right)_{\theta,t} + \left(\frac{\partial \mathcal{M}(\dot{\theta})}{\partial \theta}\right)_{q,t} \;\; = \;\; 0 \tag{B9}$$

where $m = \mathcal{M}(1)$ and thickness-weighted area integration operator $\mathcal{M}$ is defined as

$$485 \quad \mathcal{M}(*) = \int\!\!\!\int\limits_{q \leq q_0} (*)\sigma dA \;\; = \;\; \int\limits_{q^* \leq q} dq^* \oint\limits_{q*} (*)\frac{\sigma ds}{|\nabla_\theta q^*|} \tag{B10}$$

Substituting $\dot{q} = F$ and expand $\mathcal{M}(\dot{q})$ and $\mathcal{M}(\dot{\theta})$, we obtain Eq. (3).

Next, let us derive Eq. (5). Now suppose the small change of the area $\Delta \hat{A}(t)_{\chi \geq \chi_0, \lambda \leq \lambda_0}$ enclosed by the isopleth $\chi = \chi_0$ and the circle of longitude $\lambda_0$. Let $\Gamma_q$ and $\Gamma_l$ respectively be the isopleth and the circle of longitude consisting the border of the area as shown in Fig. (B1). Equation (B1) is modified as follows,

The integral of the first term on the left hand side is performed over the area $\Delta \hat{A}(t)_{\chi \geq \chi_0, \lambda \leq \lambda_0}$, and the line integral of the second term is performed only over $\Gamma_q$, as $\Gamma_l$ is constant with time. The first term can be rewritten as follows, using the Gauss'theorem for $\Gamma_q$ and $\Gamma_l$,

$$\int\limits_{\chi \geq \chi_0, \lambda \leq \lambda_0} \Delta\sigma dA \;\; = \;\; \Delta t \left[ -\int\limits_{\chi \geq \chi_0} \nabla_\theta \cdot (\sigma\boldsymbol{v}) dA - \int\limits_{\chi \geq \chi_0} \frac{\partial}{\partial \theta}(\sigma\dot{\theta}) dA \right] \tag{B11}$$

$$\;\; = \;\; \Delta t \left[ \int\limits_{\Gamma_q} \boldsymbol{v} \cdot \frac{\nabla_\theta \chi}{|\nabla_\theta \chi|} \sigma ds + \int\limits_{\Gamma_l} u\sigma ds - \int\limits_{\chi \geq \chi_0} \frac{\partial}{\partial \theta}(\sigma\dot{\theta}) dA \right] \tag{B12}$$

Then we obtain Eq. (5) with the additional term $\hat{F}(\lambda)$ defined as follows. The integral is performed over the longitude circle $\lambda_0$ consisting the border of the area.

$$\hat{F}(\lambda) \quad = \int\limits_{q \leq q_0, \lambda_0} u\sigma ds = \int\limits_{q \leq q_0, \lambda_0} u\sigma R d\phi \tag{B13}$$

*Author contributions.* AA conducted the data analysis. AA and KS contributed to the discussion and the writing of the paper.

*Competing interests.* The authors declare that they have no conflict of interest.

*Acknowledgements.* The authors thank two anonymous reviewers for their constructive comments. This study is supported by the Japan Science and Technology Agency CREST program (JPMJCR 1663) and the Japan Society for the Promotion of Science (JSPS) Grant-in-Aid Scientific Research (A) 25247075 program. All of the figures were prepared using fortran DCL developed by GFD Dennnou Club (https://www.gfd-dennou.org/index.html.en).

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

|  | $\theta$ | PV | domain | period |
|---|---|---|---|---|
| Randel and Park (2006) | 360 K | 0.93 PVU | 20–40° N | 2003 |
|  |  | (1.5 PVU as MPV) | 20–140° E | May–Sep |
| Garny and Randel (2013) | 360 K | 0.3 PVU | 15–45° N | 2005–2009 |
|  |  |  | 0–180° E | May–Sep |
| Ploeger et al. (2015) | 380 K | varies with time | 10–60° N | 2011–2013 |
|  |  | 2.6–4.4 PVU | 10° W–160° E | 20 Jun – 20 Aug |
| This study | 360 K | 0.5 PVU | 10–50° N | 1979–2016 |
|  | 370 K | 2.0 PVU | 10° W–160° E | Jun–Aug |
|  | 380 K | 3.5 PVU |  |  |

**Table 1.** Overview of the method of this study and previous studies which define the AMA based on the area enclosed by PV contours.

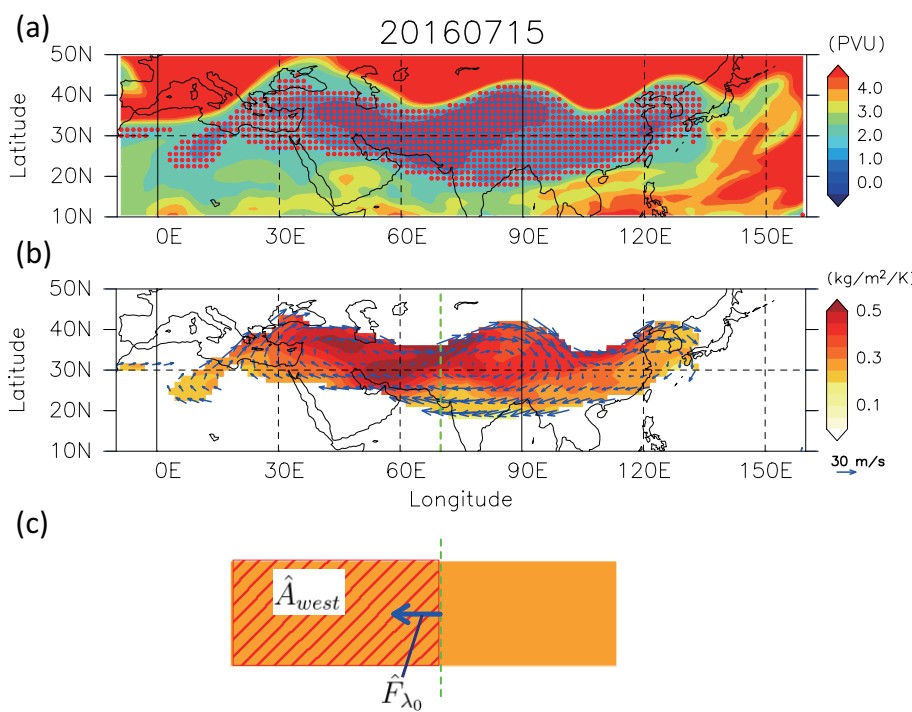

**Figure 1.** (a) Daily mean PV map at the 370 K isentropic level. The area of PV values below 2 PVU, defined as the area inside the AMA, is hatched by red dots. (b) Daily mean $\sigma$ and horizontal wind. Values are shown only for the area inside the AMA. (c) A schematic description of $\hat{A}_{\text{west}}$ and flux $\hat{F}_{\lambda_0}$. The reference longitude $\lambda_0$ is set to $70^\circ$ E in this example.

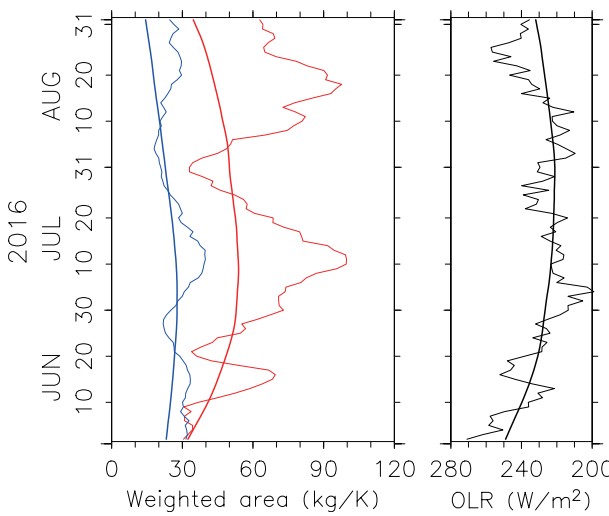

**Figure 2.** Daily thickness-weighted total low PV area defined at 360 K (red curves) and 370 K (blue curves), and the OLR averaged over 15° N – 30° N, 60° E–120° E, from June to August. Thin curves represent the daily data for 2016 and thick curves are for the 31-day filtered climatology.

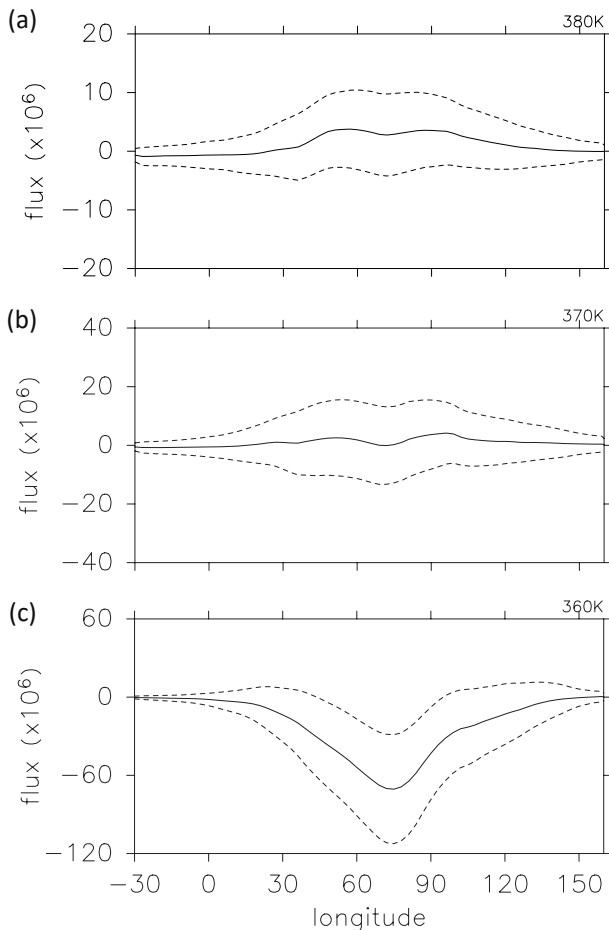

**Figure 3.** Climatological mean zonal flux of thickness-weighted low PV area as a function of longitude, calculated at the (a) 380 K, (b) 370 K, and (c) 360 K isentropic levels. The unit of the flux is $\mathrm{kg \cdot K^{-1} \cdot s^{-1}}$. Solid and broken curves correspond to mean values and ranges of standard deviations.

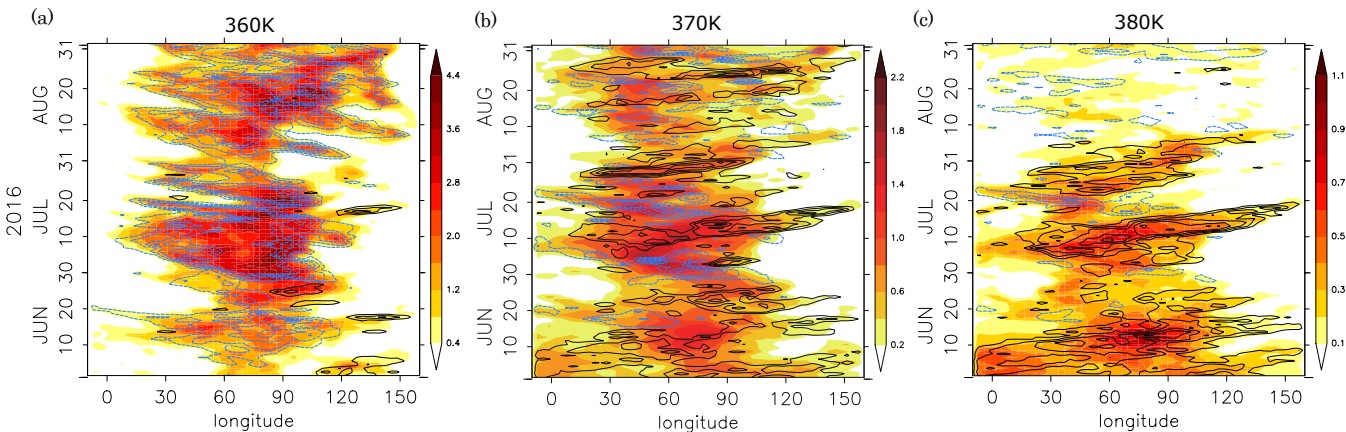

**Figure 4.** Time-longitude cross section of the air inside the AMA, defined based on the isentropic PV map at the (a) 360 K , (b) 370 K , and (c) 380 K levels. Color shadings show the thickness-weighted low PV area ($10^{12}$kg $\cdot$ K$^{-1}$) calculated for each longitude grid (1.5 degree resolution). Black solid and blue dashed contours respectively show the positive and negative zonal flux of thickness-weighted low PV area. The contour interval is $4.0, 1.5, \mathrm{and} 1.0 \times 10^7$ kg K$^{-1}$s$^{-1}$ in (a), (b), and (c), respectively.

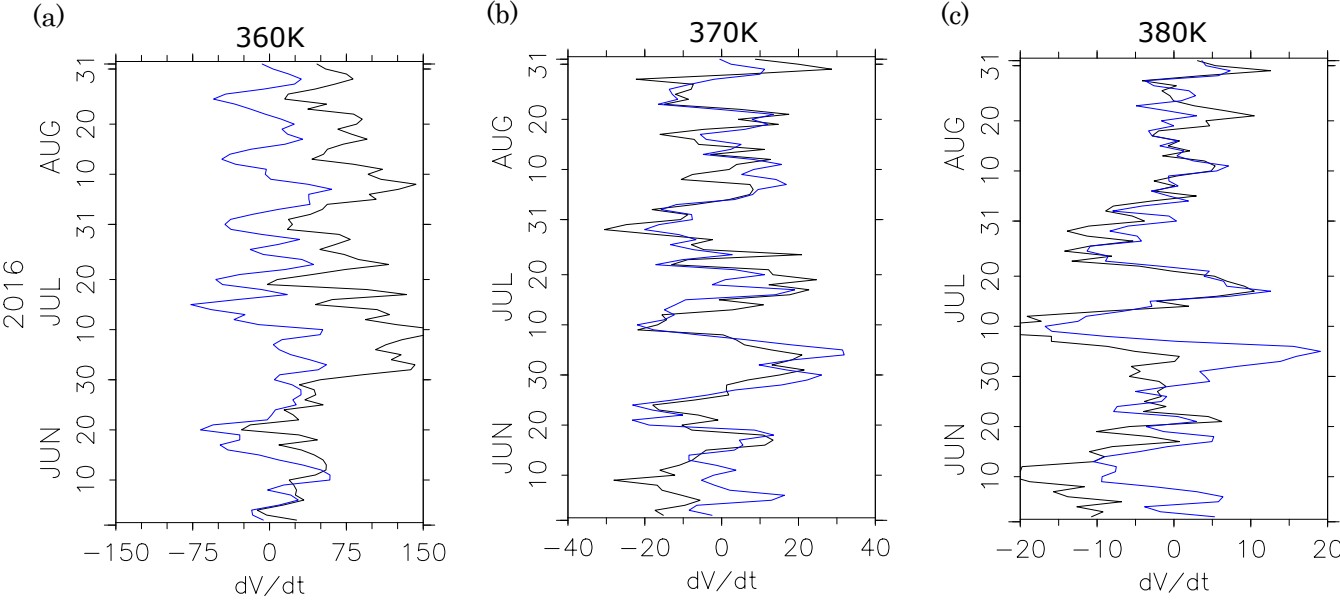

**Figure 5.** The tendency of the partial thickness-weighted anticyclone area $\frac{d}{dt}\hat{A}_{\text{west}}$ (blue: the left hand side of Eq. 6) and westward zonal flux entering the area $-\hat{F}$ at $60°$ E (black: the first term on the right hand side) calculated at the (a) 360 K , (b) 370 K , and (c) 380 K levels. The unit of the horizontal scale is $10^6$ kg $\cdot$ K$^{-1}$s$^{-1}$.

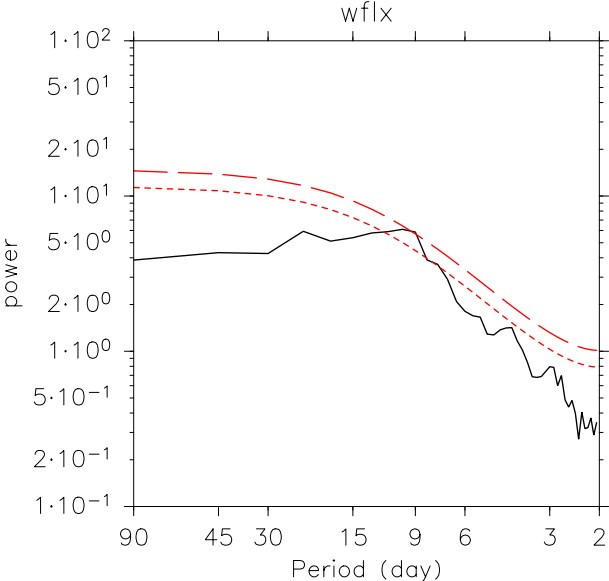

**Figure 6.** Power spectrum of the longitudinal flux of thickness-weighted low-PV area at 60° E at the 370K level, calculated for June–August and averaged over 1979–2016. Broken red curves represent 95 and 99 confidence levels.

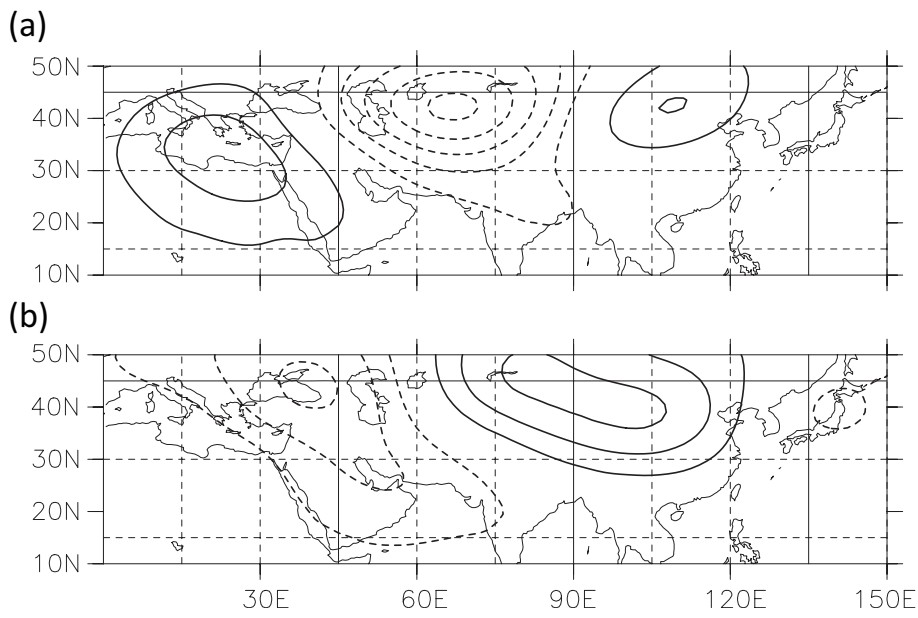

**Figure 7.** Spatial structure of (a) EOF 1 and (b) EOF 2 modes in geopotential height. The contour interval is 6 m. Zero contours are suppressed.

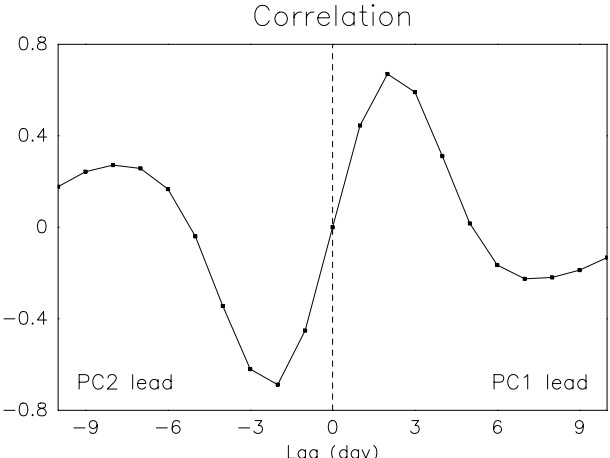

**Figure 8.** Lag-correlation in the unit of day between the principle component corresponding to the first and second EOF modes.

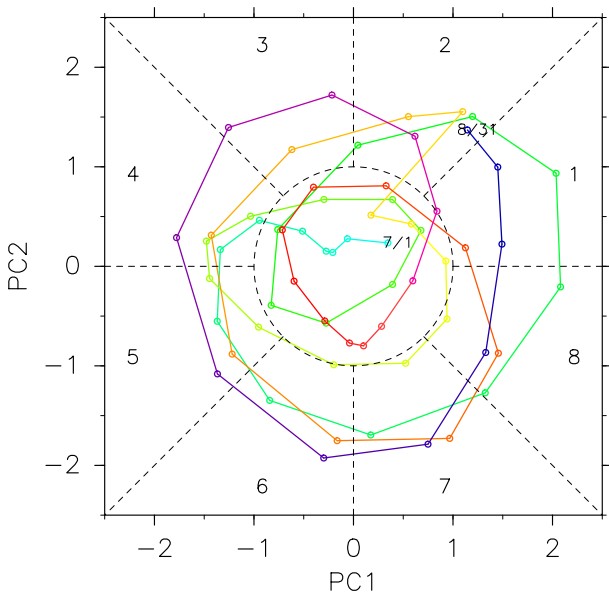

**Figure 9.** The definition of eight phases based on normalized two leading principal components. The daily phase progress in the year 2016 is shown as an example.

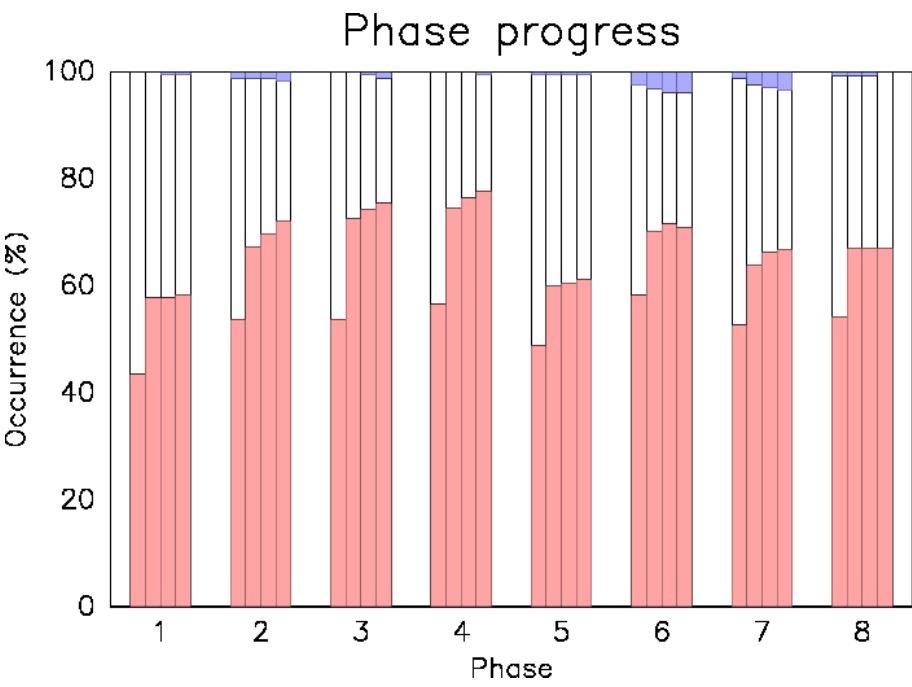

**Figure 10.** Percentage of forward (red) and backward (blue) phase progress starting from each phase. Four bars from left to right in each column correspond to time lags of 1 to 4 days.

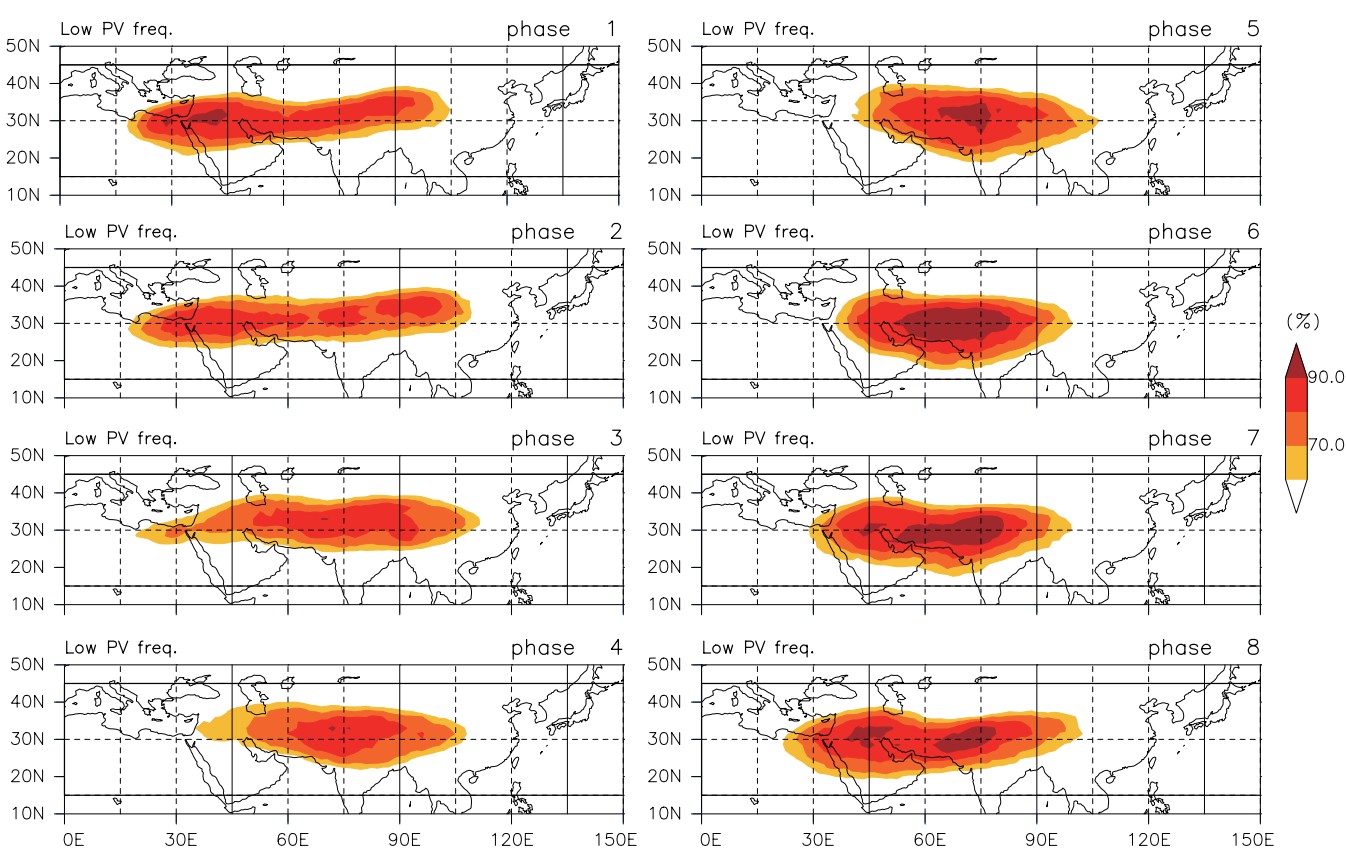

**Figure 11.** Percentage of PV value lower than the reference value of 2.0 PVU at the 370 K isentropic level for each phase.

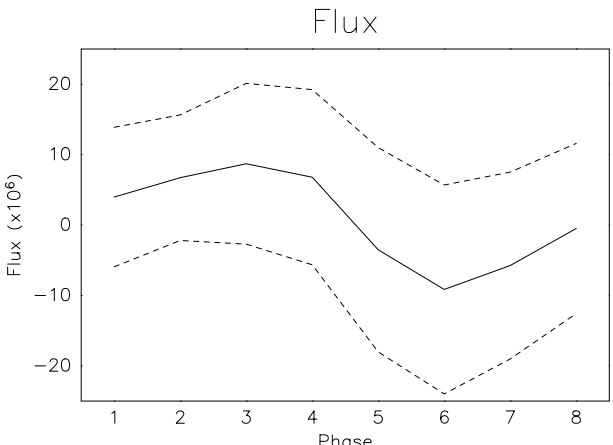

**Figure 12.** Mean value (solid line) and the range of standard deviation (broken lines) of the zonal flux of thickness-weighted low PV air $(\mathrm{Kg}\ \mathrm{K}^{-1}\mathrm{s}^{-1})$ at 60 $^\circ$ E calculated at the 370 K level for each of 8 phases.

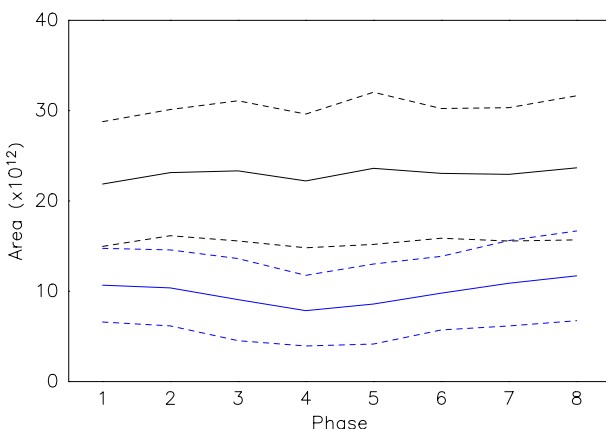

**Figure 13.** Mean values (solid curves) and the ranges of standard deviation (broken curves) of the area of thickness-weighted low PV air $(\mathrm{Kg\,K^{-1}})$ calculated at the 370 K level for each of the 8 phases. Black and blue curves correspond to the total area and the partial area westward of 60 ° E, respectively.

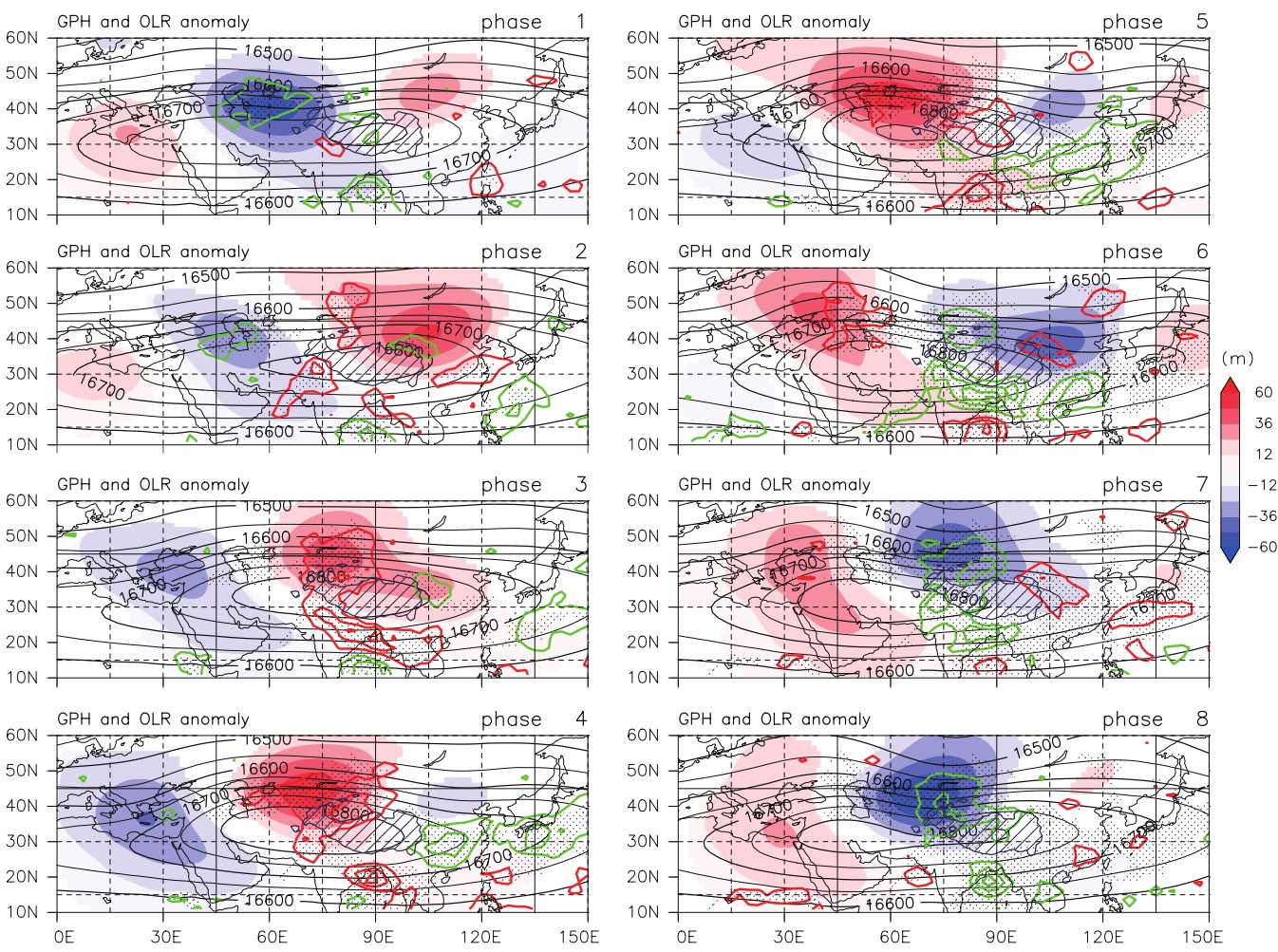

**Figure 14.** Composite maps of geopotential height at 100 hPa level (black contour) and its anomaly (shade), and the OLR anomaly (green and red contour) for each phase. Anomalies are calculated as deviations from climatology. The contour interval for geopotential height is 50 m. The contour interval for OLR anomaly is 5 W m$^{-2}$, with green and red contours respectively representing negative and positive anomalies. Shading of geopotential anomalies show only areas with 95% significance by a standard $t$-test. The areas of OLR anomaly with 95 % significance is hatched by dots. Areas with an elevation above 3000 m are hatched.

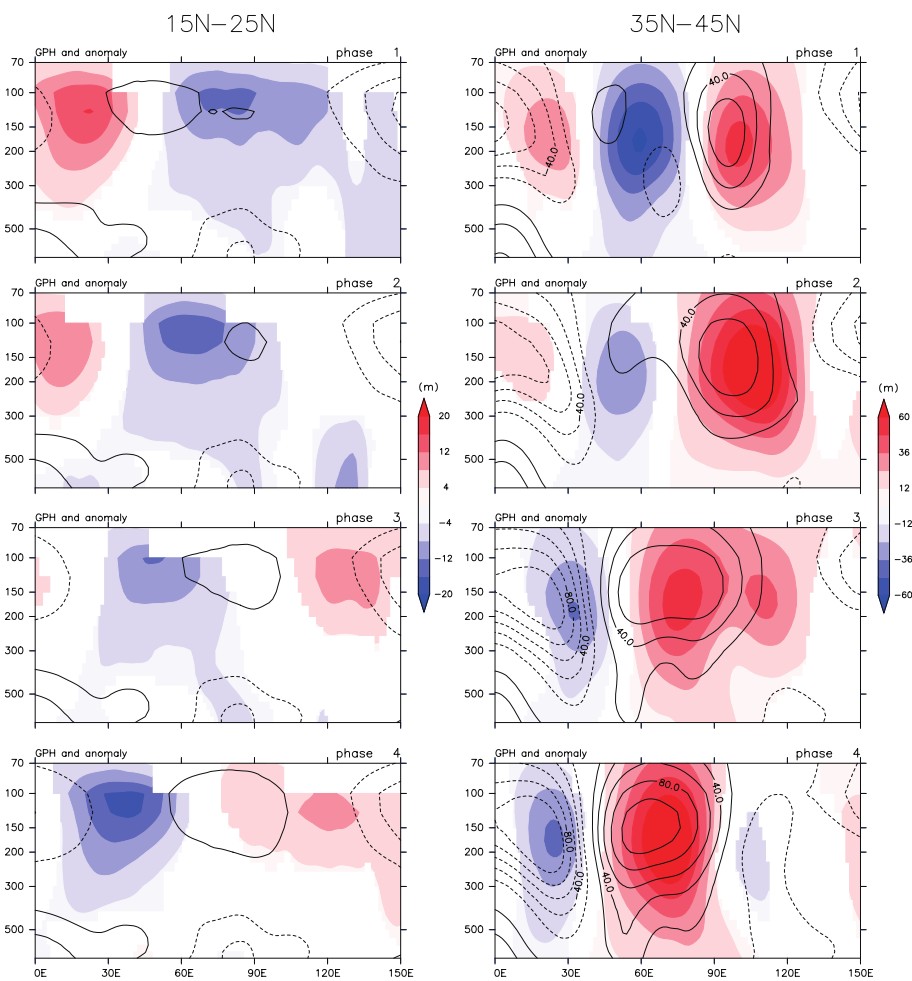

**Figure 15.** Composite of geopotential height in the longitude-pressure section. Contours show deviation from the zonal mean of the longitude sector, and its anomalies from climatology are shaded. The left and right columns respectively show maps averaged over 15° N–25° N and 35° N–45° N. Only the first four of eight phases are shown. The contour interval is 20 m. Only the areas with 95% significance, determined by standard t-test, of geopotential anomalies are shaded. Note that different color scales are used for the left and right columns.

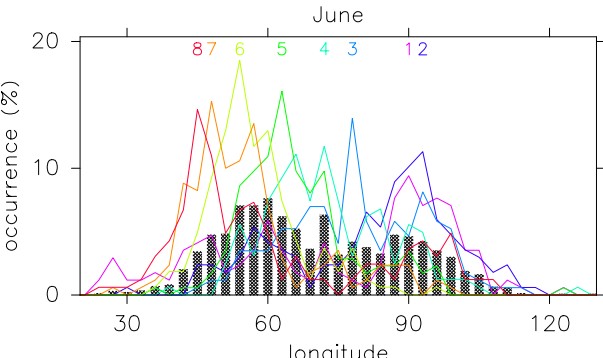

**Figure 16.** Longitudinal distribution of the occurrence frequency of the AMA center defined as the geopotential maximum at the 100 hPa pressure level. Hatched bars show the total average. Lines show partial averages of the data assigned for each phase, as denoted by numbers of the corresponding colors.

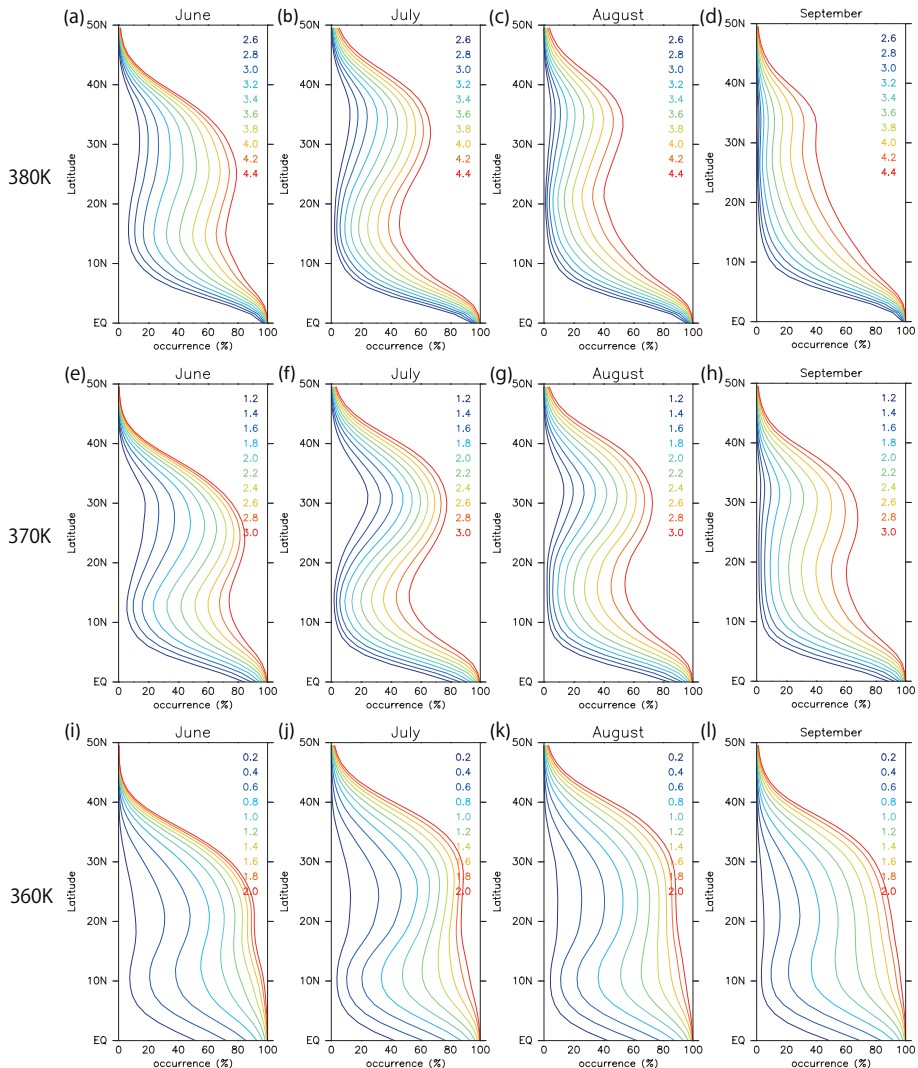

**Figure A1.** Percentage of the occurrence of grid points where PV is below the reference value between $0°$ E–$160°$ E, as functions of latitude, calculated using ERA-Interim data from 1979 to 2016. Different line colors correspond to different reference PV values which are shown in the top-right of each figure.

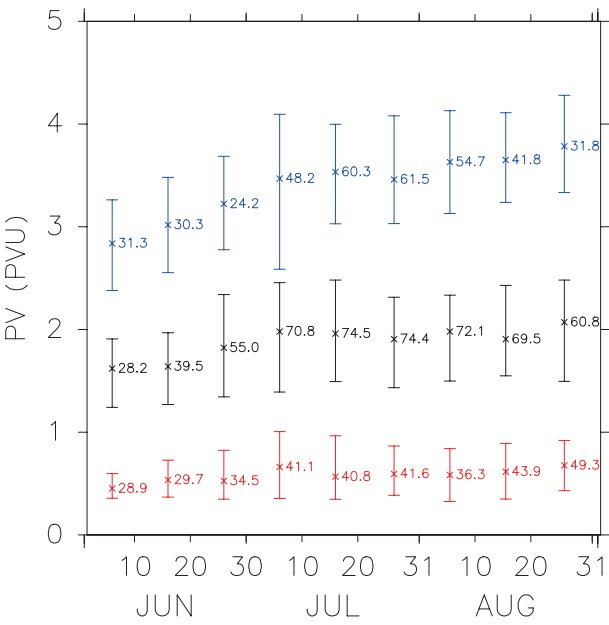

**Figure A2.** Statistics for the daily barrier PV values, determined objectively by the method of Ploeger et al. (2015), using ERA-Interim reanalysis data from 1979 to 2016. The horizontal axis is the time of year, with nine time ranges corresponding to the early, middle, and late periods of each month between June and August. Red, black and blue lines respectively show the results for 360, 370, and 380 K levels. Vertical lines show the range of PV values between 15th and 85th percentiles, with crosses in the middle corresponding to the median values. The numbers next to the crosses represent corresponding percentages of the cases when the barrier is successfully determined.

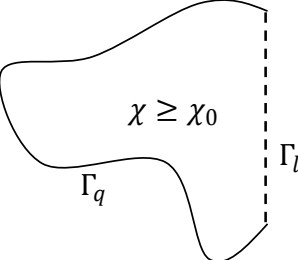

**Figure B1.** An example of the area enclosed by the isopleth of $\chi$ and a specific longitude. $\Gamma_q$ is the part of the isopleth $\chi = \chi_0$. $\Gamma_l$ is the part of the circle of longitude $\lambda = \lambda_0$.