# Peer review of "Characterizing quasi-biweekly variability of the Asian monsoon anticyclone using potential vorticity and large-scale geopotential height field"

_Atmospheric Chemistry and Physics, 2020_

## Referee Comment (RC1) · Anonymous Referee #1 · 9 Jul 2020

**Review of Amemiya and Sato, 2020**

July 9, 2020

The study by Amemiya and Sato addresses the characterization of the variability of the Asian monsoon upper tropospheric anticyclone on sub-monthly timescales. The focus is on describing the east-west oscillations of the anticyclone on quasi-biweekly timescales, and assessing whether this movement is driven by variability in convective forcing, or whether the total area of the anticyclone (in terms of weighted low PV area) is conserved during this process. They show that both the analysis of variability of the low PV area, as well as analysis based on variability modes (EOFs) of geopotential height anomalies reveal the biweekly oscillation of the anticyclone, and argue that the oscillation is mostly associated with "passive advection" (i.e., not with variable forcing).

The paper addresses open research questions on the anticyclone variability, and used novel and promising techniques. However, I have a number of concerns regarding the derivation of the methodology, and regarding the interpretation of the results. After those issues are properly addressed, the paper will surely be an important contribution to the literature on the variability of the anticyclone.

**1 Major comments**

1. My first major concern is on the way, the methodology of the "weighted low PV area" is introduced and motivated. While the PV-area conservation equation (Equ. 1) is widely used, the low PV area weighted by the thickness (isentropic density sigma) is introduced in this study (at least to my knowledge). First of all, the weighted low PV area ($\hat{A}$) should be properly defined, e.g. as:

$$\hat{A} = \int_\lambda \int_\phi \sigma r^2 cos(\phi) d\phi d\lambda \tag{1}$$

Most importantly, I would like to see the derivation of the equation for the weighted low PV area $\hat{A}$, i.e. Equ. (3) in the paper. I don't see that it is straight forward to obtain this equation (if so, please state the necessary steps), so a proper derivation (possibly in the Appendix) would be important. Moreover, the derivation of the equation for only the western part of the PV area (Equ. (4)) could also be explained better, e.g. to elucidate the emergence of the Flux term. Is the diabatic term (term 2 on right hand side of Equ. (4)) indeed integrated over the whole low PV area, or only over the western part, as I would assume? If sou, please modify the equation to make this clear.

Next to the proper definition and derivation, I would appreciate a deeper physical reasoning for the weighting of the PV area. If I understand it correctly, the relevant effect

here is that, given the conservation of PV for a given air mass, a vertical (i.e, in theta) compression of this air mass leads to a larger horizontal extend, i.e. an increase in area. Thus, by applying this scaling you essentially move to an equivalent 2-d representation of the air parcel, and $\hat{A}$ is the area of this equivalent 2-d air parcel. Maybe the addition of a simple sketch would help the reader to get a better understanding of the meaning of the scaling.

A related question on the scaled versus non-scaled PV area: according to your equations, the divergence term vanishes for the density-scaled area equation. If my above understanding of the weighted-PV area is correct, this would imply, that the role of the divergence term merely is to deform the air parcel. However, as you mention, in earlier work it was shown that the divergence term in the conventional PV area equation is closely related to the diabatic term, and indeed maximizes in the regions of low OLR (as indication for convection). Following this finding, the interpretation would be that the divergence (i.e., convective outflow) induces the low PV, and not merely deforms the air parcel. Is this effect completely incorporated in the diabatic term in the weighted low PV equation, or how can I understand the physical meaning of the terms?

2. The period of the "quasi-biweekly" oscillation is not clearly quantified until rather late in the paper, namely with the power spectrum in Fig. 6. Rather, the motivation for those time-scales is given by visual analysis of the timeseries of the weighted low PV area in Fig. 2. I'd encourage the authors to move the power spectrum to an earlier point in the paper, and compare it to the power spectrum of the total weighted low PV area (see also specific comment below). Further, the peak in the power spectrum of the fluxes (Fig. 6) is rather broad (with a plateau-like peak between 20 to 9 days), does this range of time-scales still correspond to the "quasi-biweekly" oscillation?

3. My third major comment is on the interpretation of the results, mainly the overarching question of the nature of the quasi-biweekly oscillation as "passive advection" versus reflecting variability in sources (i.e., convective forcing) or sinks (e.g., the actual shedding of air from the anticyclone). This comes down to the question, whether the total low PV area is conserved during the oscillations (as is shown for the life cycle analysis in Fig. 13), or, whether the total flux across the defined boundary at 60 E explains the in- and decrease of low PV area to the west and east of the boundary. The latter is indicated to be the case by the close match of timeseries of area change and fluxes in Fig. 5. However, I wonder whether this analysis couldn't be made more quantitative by repeating it for longer time series (multiple years), and actually quantifying to which degree the total western PV area is explained by the flux (for both increasing and decreasing areas, i.e., positive versus negative values of $d\hat{A}/dt$ and the flux $F$ ). Also, I wonder to which degree the eastern part of low PV area is explained by the flux term. As the diabatic source is located east of 60 E, it maybe not surprising that the western part is explained by the fluxes from the east. In general, also the location of the boundary at 60 E could be varied to test the sensitivity of the results on the choice of the boundary longitude.

Moreover, the results mentioned above are all valid for the 370 K level (if I'm not mistaken, I found it hard to identify which level the analysis is performed on at many places, see specific comment below). According to your Fig.3, in which the fluxes of low PV area across the 60 E line are shown for both 370 K and 360 K, the fluxes behave rather different at the two levels: At 370 K, the flux is close to zero in the mean (indicating back- and

forth advection), while at 360 K, the flux is clearly eastward, which is in accordance with the "source" of low PV (i.e. convection) being mostly located east of 60E. Therefore, I wonder in how far the result on the "passive advection" of low PV air is valid also for the 360 K level (which does not lie well above the main convective outflow and heating level, as does the 370 K level).

4. Another major comment I have is on the interpretation of the OLR anomalies during the oscillation life cycle (as presented in Fig. 14). I can identify from the figure a clear signal in the OLR anomalies, with westward propagating negative OLR anomalies during phase 5 to 8, and positive OLR anomalies during phase 2 to 4. Thus, this might suggest that variability in the forcing of the anticyclone does play a role on those time-scales after all. This possibility is indeed phrased in the summary (lines 329-330), but this is a bit controversial to what is stated earlier (e.g. lines 302 to 303). On the other hand, the consistent OLR anomalies could also indicate that the quasi-periodic circulation anomalies of the anticyclone influence the occurrence of convection. This result might be consistent with the troposphere-deep circulation anomalies at 35-45 N, as shown in Fig. 15. This possible implication is discussed in the summary (lines 335 onwards), but I'd suggest that you could add here, that the OLR anomalies in the "life cycle" also show indications in this direction.

**2  Specific / minor comments**

- title: Change to "geopotential height fields". In my opinion, the latter half of the title (".. using PV and geopotential...) could also be skipped, but this is a matter of taste, so I leave it to the authors to decide.

- line 24: "to be dominant": consider rephrasing to "to be the dominant transport process"

- line 48-50: I wouldn't agree in that the paper by Nuetzel et al showed that the bimodality is a robust feature. Indeed, they showed that the bi-modality is very prominent only in older (NCEP) reanalysis data sets.

- line 113: This sentence makes it sound as if the low PV area in the anticyclone is usually conserved, but just not strictly, because of the "forcing processes such as deep convection". Diabatic heating and associated outflow from deep convection (divergent motion) is THE forcing process of the anticyclone, if it wasn't for that, there would be no low PV area to start off with. Therefore, I find this formulation a little weird. Please rephrase it to make the role of deep convection on forcing low PV more clear.

- line 130 / Equ. 2: On a similar note as major comment 1, a definition for the longitude-dependent quantity $L(\lambda, t)$ along the lines suggested for $\hat{A}$ could be given (i.e. as integral over $\phi$).

- line 171: here, the authors state that the analysis of the timeseries in the preceding subsection "confirmed" that the dominant timescale of variability is the "quasi-biweekly" timescale. However, in the daily timeseries presented in Fig. 2, a monthly period is predominant, and the quasi-biweekly timescale is, if at all, only to be guessed "by eye". So either you have to weaken the statement here (e.g. indicates that quasi-biweekly

variability can be identified from the timeseries) or make the analysis more quantitative (see also major comment 2).

- line 186: For 360 K, the flux is negative around its minimum, even within the range given by the standard deviation. So this argument holds only for Fig 3a (370 K), right?

- line 193/194: "fixed latitude range in the southern part of the AMA" - in Garny et al, the total low PV area within 15-45 N was shown (see their Fig. 6), so this is not really a fixed latitude range, and neither only the southern part? The main difference is, apart from the slightly larger latitude range here, rather the weighted versus non-weighted low PV area, and the addition of the fluxes.

- line 197: why only "as far as PV conservation holds"? Is the flux not valid if the the diabatic term is not equal zero? This would be worrisome for the whole analysis of the paper.

- line 209: "dominant period of the variability in total low PV area" - actually show the power spectrum of the low PV area (see also comment on line 171, and major comment 2)?

- line 219: Do you mean to say that the timeseries filtered with a band pass filter within 5-20 days periods? Please specify.

- line 232: "zonally averaged total perturbation variance": Do you mean the variance in terms of anomalies at each longitude, and then this variance is zonally averaged?

- line 233: the studies mentioned here rather analyzed tele-connections to the mid-latitudes than variability of the anticyclone itself, correct? So maybe it is not surprising that they find different pattern? ( Also, correlation to the time-series "at a point at a midlatitude" is a bit vague - please clarify).

- I find the EOF analysis, and the PC lag analysis and life cycle a great approach to characterize the variability. Possibly, adding actual data points to Figure 9 to see the progression of the phases would be beneficial?

- Fig. 11: agree that there is clear westward extension from phase 5 to 8, but does the low PV area "move back" to the east from phase 1 to 4, or is it shed? From the extend and strength of the low PV occurrence, it seems like the total PV area decrease over those phases. Related, is the total integral over the westward flux (in Fig. 12) equal to the sum of the eastward Flux over all phases? This would prove this point, and I guess it has to be the case, given that the total area appears to remain rather constant according to Fig. 13.

- Fig. 15: Not sure what the difference of black contours and color shading is - deviations from zonal mean versus anomalies from this deviation?

- lines 281-282: I'm not sure I understand the statement on the role of the subtropical jet on the deep gph anomalies. Please either remove, or add explanation/ citation for this statement.

- line 307: would you consider 30% of variability from both the 1st and 2nd EOF together the "dominant variability"?

- line 305/306: here you state that the west/east-ward flux of low PV is consistent with "eddy shedding", which could be true, but this notation implies that the eddies are at least partly "shed" from the Anticyclone, while your analysis seems to suggests that the total area is conserved during the quasi-biweekly variability, i.e. no actual "shedding" occurs. In Fig.3 b), where you show that the total eastward flux at 360 K is negative, does that imply actual westward shedding?

- general: State in all Figure caption, and possibly more often in the text, at which level the analysis is performed on! (I found this information to be rather hidden).

**3   Typos / technical**

- line 165: "pointed out by..." (add "out")

- line 220: dividing "by" their standard deviation (insert "by" )

- line 225: "longer months": change to "longer period" ?

- line 275: "rest phases": change to "for the rest of the phases" ?

- Fig. 15: Title and legend should say 15-25N and 35-45N rather then E.

- line 409: "noize": change to "noise"

- line 420: "persentage": change to "percentage"

- Fig. 4: which level?

- Fig. 13: I would suggest to change/remove the heading "A_west", as not only "A_west" is shown

---

## Referee Comment (RC2) · Anonymous Referee #2 · 13 Jul 2020

The manuscript is an interesting study of the variability of the Asian monsoon anticyclone (AMA) that brings new results on the quasi-biweekly variability and should eventually be published. There are, however a number of major and minor points that deserve further work from the authors

Major points

It is unclear whether it is a major or minor point but the basic equation (1) which is taken from Garny and Randel (2013) is technically wrong as it is presented. The integrand

dS in the first member of the r.h.s. is not a line element but a line element divided by the modulus of the horizontal gradient of the PV. This is stated by Garny and Randel and, otherwise, the equation is not even dimensionally homogeneous. I hope that this detail has not been missed by the authors and that the error was only introduced during writing but it is quite worrying.

The main claim of the manuscript is that the oscillations are mostly of dynamical origin and reversible, and that forcing by convection and dissipation are not involved. This is quite opposite to conclusions of previous works and also to Wei et al. (2019, doi:10.1029/2019GL086180) and references herein which is another stream of research that should be quoted and discussed. Another relevant work that studies PV fluxes on isentropic surfaces is Ortega et al. (2018, doi: 10.1002/qj.3261) which is also missing in the reference and should be used to compare the results of the manuscript.

The manuscript focuses on the zonal mass flux of air with low PV and finds that the mean flux at 370K oscillates around zero over the range of latitudes of the AMA. This is basically the main result. However, this does not mean that there is no zonal flux of PV. It is clear from fig.11 that during the phase of eastward flux at 60E, the air carries less negative PV that during the phase of westward flux. Therefore the zero mean mass flux does not rule out a non zero mean PV flux, where negative PV is created on the east by convection and dispersed and lost to the background on the west by vortex shedding within a biweekly cycle. It is useful to notice that the circulation time around the anticyclone and its erosion rate are also of about two weeks (Legras and Bucci, 2019, doi: 10.5194/acp-2019-1075). PV is clearly not well conserved during the bi-weekly cycle.

Minor points

l. 51-53: Is it so clear that the two questions are well separated?

l. 63 "is often"

l. 141: I do not see why the divergence term disappears in this equation.

l. 145: The integral is at fixed longitude and the integrand is latitude over the range where the PV is below the threshold and F is the mass flux (rather than the movement) of low PV air across a given latitude. This is badly described and the scheme in fig.1c adds to the confusion.

l. 164: I do not see the need for a 31-day filter when the average is done over 38 years. This should be enough to scramble the phases of the AMA oscillations.

l. 173: At this stage, the evidence is only based on the visual appearance of a single year record.

l. 176: This line should refer to eq. (5) if this is what is shown.

l. 181: This line makes me worrying whether the total heating, including latent heating, is accounted as it should in this work or whether only radiative heating is used. At 370K, it is however correct to assume that radiative heating dominates.

l. 190: I asume that the results are shown on the 370 K surface but this should be stated. It is very difficult to distinguish the blue and red contours in fig. 4. The text mention that fig.4 shows the mass weighted length that should have dimension kg/(K x m) and the caption says that it is a weighted area with dimension kg/K. Please clarify. Provide a definition for this weighted area that depends on the longitude and discard L if is not used.

In the sequel, no PV diagnostic is shown on other surfaces than 370K. The choice of 370K is justified in the appendix on the basis of the best definition of AMA in terms of PV but is would nevertheless be interesting to look at over surfaces. 380 K was privileged in Ploeger et al. (2015) and 360 K is closer to the level where convective detrainement is the strongest. It is also where the mean eastward and westward branches of the AMA are maximum and where the isentropic divergence is maximum.

l. 197: I do not see why PV conservation is invoked here. It is clear that PV is not well

conserved here (see fig.11).

l. 209 and Fig. 6: How should we interpret the significance curves on Fig. 6? The peak is not that strong and shows there is a plateau in the spectrum intensity between 9 days and 25 days. 9 days is more a cutoff period than a dominating period.

Sect. 4 It is a bit surprising that the study switches here to the geopotential on the 100 hPa surface. Having done all the work to interpolate basic variables on isentropic surfaces would have made easy to calculate the Montgomery potential on such surfaces. Basically, the results would not have been very different but this would have been more consistent, especially because isentropic and isobaric surfaces may differ quite significantly in the Asian monsoon region.

As the authors are looking for a cycle, they should have considered the MSSA method which is particularly well suited (Ghil et al, 2012, doi:10.1029/2001RG000092) and would have saved time and space.

l. 220 "dividing by their"

l.220 Why the square root of grid area and not the area in the weight?

l. 240: It is quite difficult to understand fig. 9 which is introduced in a section where PV plays no role. Please improve the caption such that it makes sense when the reader is at line 240 in the text.

Sect. 4.2 The first paragraph concludes that variability is determined by internal inviscid and adiabatic dynamics but the second paragraph shows there is a pattern of convection associated with the oscillation which somewhat contradicts the first paragraph if we admit that convection does not only react passively but generates a forcing. The authors do not attempt to provide a balanced view and just discard the convective influence in this section and in the conclusions l.323-330.

---

## Author Comment (AC1) · 24 Aug 2020

We greatly appreciate the reviewer's invaluable and constructive comments. We have revised our manuscript in response to your and the other reviewer's comments. Responses to each of the major and minor comments are written below.

1. Major comments

> My first major concern is on the way, the methodology of the "weighted

low PV area" is introduced and motivated. While the PV-area conservation equation (Equ. 1) is widely used, the low PV area weighted by the thickness (isentropic density sigma) is introduced in this study (at least to my knowledge). First of all, the weighted low PV area ( $\hat{A}$ ) should be properly defined.

As you suggested, we have added the definition of the PV area to the main text as follows.

$$\hat{A}(t)_{q \le q_0} \equiv \int_{q \le q_0} \sigma dA = \int \int \sigma r^2 \cos \phi d\lambda d\phi, \qquad (1)$$

> Most importantly, I would like to see the derivation of the equation for the weighted low PV area  $\hat{A}$ , i.e. Equ. (3) in the paper. Moreover, the derivation of the equation for only the western part of the PV area (Equ. (4)) could also be explained better, e.g. to elucidate the emergence of the Flux term. Is the diabatic term (term 2 on right hand side of Equ. (4)) indeed integrated over the whole low PV area, or only over the western part, as I would assume? If sou, please modify the equation to make this clear.

We also agree that Eqs. 3 and 4 need proper derivation, as also suggested by the other reviewer. In addition, as the other reviewer pointed out, the equation for unweighted low PV area, Eq.1, was not properly described in the original manuscript. The multiplicative factor for line element ( $|\nabla_{\theta}q|^{-1}$ ) was missing, while it was properly described in the reference article (Garny and Randel, 2013). Eqs. 3 and 4 need to be revised in a similar way. Moreover, we have noticed that both of those equations have another missing term regarding diabatic heating. We are sorry for the inaccurate description of the original manuscript. The correct forms of Eqs. 3 and 4 are as follows. The derivation of Eqs. 3 and 4 has been added to Appendix B of the revised manuscript and also at the bottom of this reply.
**ACPD**

Interactive

comment

$$\begin{array}{lcl} \displaystyle \frac{d}{dt} \hat{A}_{\mathsf{tot}}(t) & = & \displaystyle \oint_{q=q_0} \left( -q \frac{\partial \dot{\theta}}{\partial \theta} + \dot{\theta} \frac{\partial q}{\partial \theta} \right) \ \displaystyle \frac{\sigma dS}{|\nabla_{\theta} q|} - \int_{q \leq q_0} \frac{\partial}{\partial \theta} (\sigma \dot{\theta}) dA \\ & + & (\mathsf{unresolved term}) \end{array}$$

$$\frac{d}{dt}\hat{A}_{\mathsf{west}}(t) = -\hat{F}(\lambda_0) + \oint_{q=q_0,\lambda \le \lambda_0} \left( -q\frac{\partial\dot{\theta}}{\partial\theta} + \dot{\theta}\frac{\partial q}{\partial\theta} \right) \sigma dS - \int_{q \le q_0,\lambda \le \lambda_0} \frac{\partial}{\partial\theta}(\sigma\dot{\theta}) + (\mathsf{unresolved term}) \tag{3}$$

> Next to the proper definition and derivation, I would appreciate a deeper physical reasoning for the weighting of the PV area. If I understand it correctly, the relevant effect here is that, given the conservation of PV for a given air mass, a vertical (i.e, in theta) compression of this air mass leads to a larger horizontal extend, i.e. an increase in area. Thus, by applying this scaling you essentially move to an equivalent 2-d representation of the air parcel, and  $\hat{A}$  is the area of this equivalent 2-d air parcel. Maybe the addition of a simple sketch would help the reader to get a better understanding of the meaning of the scaling.

We appreciate the suggestion. As you pointed out, the weighting of the PV area is related to tracking an air parcel on 2-d surface. We have added the explanation about it at lines 151-160 in Section 2 of the revised manuscript.

> A related question on the scaled versus non-scaled PV area: according to your equations, the divergence term vanishes for the density-scaled area (2)

equation. If my above understanding of the weighted-PV area is correct, this would imply, that the role of the divergence term merely is to deform the air parcel. However, as you mention, in earlier work it was shown that the divergence term in the conventional PV area equation is closely related to the diabatic term, and indeed maximizes in the regions of low OLR (as indication for convection). Following this finding, the interpretation would be that the divergence (i.e., convective outflow) induces the low PV, and not merely deforms the air parcel. Is this effect completely incorporated in the diabatic term in the weighted low PV equation, or how can I understand the physical meaning of the terms?

It is the diabatic term that induces low PV, not directly the divergent term. The divergent term is likely to follow the diabatic forcing by deep convection, resulting in high correlation between it and low OLR, but the divergent term without nonconservative forcing does have nothing to do with the weighted low PV area. The revision we made in lines 151-160 in Section 2 in the revised manuscript includes this explanation.

> The period of the "quasi-biweekly" oscillation is not clearly quantified until rather late in the paper, namely with the power spectrum in Fig. 6. Rather, the motivation for those time-scales is given by visual analysis of the timeseries of the weighted low PV area in Fig. 2. I'd encourage the authors to move the power spectrum to an earlier point in the paper, and compare it to the power spectrum of the total weighted low PV area (see also specific comment below). Further, the peak in the power spectrum of the fluxes (Fig. 6) is rather broad (with a plateau-like peak between 20 to 9 days), does this range of time-scales still correspond to the "quasi-biweekly" oscillation?

We did not mean to state that "quasi-biweekly" timescale is dominant here yet. Rather, section 3.1 examplifies that the total weighted PV area  $\hat{A}_{tot}(t)$  shows over-30 day vari-

**ACPD**
ability dominates as already mentioned in earlier studies and there are also secondary signals with a shorter time scale. In this sense, we consider the sentense 'The analysis regarding the total area so far confirmed ... ' was misleading. We have revised the first paragraph of section 3.2 to make it clear. Also, we agree that the peak in Fig. 6 does not show clear characteristic time scale of 'quasi-biweekly'. There is broad range of possible characteristic time scale centered around quasi-biweekly. We have revised lines 226-228 in the revised manuscript.

> My third major comment is on the interpretation of the results, mainly the overarching question of the nature of the quasi-biweekly oscillation as "passive advection" versus reflecting variability in sources (i.e., convective forcing) or sinks (e.g., the actual shedding of air from the anticyclone). This comes down to the question, whether the total low PV area is conserved during the oscillations (as is shown for the life cycle analysis in Fig. 13), or, whether the total flux across the defined boundary at 60 E explains the in- and decrease of low PV area to the west and east of the boundary. The latter is indicated to be the case by the close match of timeseries of area change and fluxes in Fig. 5. However, I wonder whether this analysis couldn't be made more quantitative by repeating it for longer time series (multiple years), and actually quantifying to which degree the total western PV area is explained by the flux (for both increasing and decreasing areas, i.e., positive versus negative values of  $d\hat{A}/dt$  and the flux F). Also, I wonder to which degree the eastern part of low PV area is explained by the flux term. As the diabatic source is located east of 60 E, it maybe not surprising that the western part is explained by the fluxes from the east. In general, also the location of the boundary at 60 E could be varied to test the sensitivity of the results on the choice of the boundary longitude.

We have found that the close relationship between the flux term F and the low PV area
to the west of 60E is satisfied in general, also for other years. We have shown the correlation factor 0.663 between  $d\hat{A}/dt$  and the flux *F* for 1979-2016 in line 204 of the original manuscript. We consider this value shows that the correspondence between these two terms is quite significant. We do not expect the similar clear relationship for the eastern part of the anticyclone, where the diabatic forcing term has a significant role.

> Moreover, the results mentioned above are all valid for the 370 K level (if I'm not mistaken, I found it hard to identify which level the analysis is performed on at many places, see specific comment below). According to your Fig.3, in which the fluxes of low PV area across the 60 E line are shown for both 370 K and 360 K, the fluxes behave rather different at the two levels: At 370 K, the flux is close to zero in the mean (indicating back and forth advection), while at 360 K, the flux is clearly eastward, which is in accordance with the "source" of low PV (i.e. convection) being mostly located east of 60E. Therefore, I wonder in how far the result on the "passive advection" of low PV air is valid also for the 360 K level (which does not lie well above the main convective outflow and heating level, as does the 370 K level).

We have revised Figs. 4 and 5 to show the results at three different levels 360, 370, and 380 K. As expected, the contribution of the longitudinal advection to the low PV area distrubution is less clear at 360 K at 380K, as there are more significant source and sinks. The whole picture of the budget of low PV airmass is more complicated than what is seen at a single level, we consider the longitudinal ocsillatory behavior with a time scale of quasi-biweekly is one important feature, and the low PV air at 370 K is the best proxy for that feature. We have revised section 3.2 adding this discussion.

> Another major comment I have is on the interpretation of the OLR anoma-
lies during the oscillation life cycle (as presented in Fig. 14). I can identify from the figure a clear signal in the OLR anomalies, with westward propagating negative OLR anomalies during phase 5 to 8, and positive OLR anomalies during phase 2 to 4. Thus, this might suggest that variability in the forcing of the anticyclone does play a role on those time-scales after all. This possibility is indeed phrased in the summary (lines 329-330), but this is a bit controversial to what is stated earlier (e.g. lines 302 to 303). On the other hand, the consistent OLR anomalies could also indicate that the quasi-periodic circulation anomalies of the anticyclone influence the occurrence of convection. This result might be consistent with the troposphere-deep circulation anomalies at 35-45 N, as shown in Fig. 15. This possible implication is discussed in the summary (lines 335 onwards), but I'd suggest that you could add here, that the OLR anomalies in the "life cycle" also show indications in this direction.

We consider the exsitence of the significant OLR signal does not rule out the possiblity that the variability is driven by internal dynamics, then lines 329-330 and 302-303 of the original manuscript does not contradict with each other. The OLR anomaly in composite map can be just a response to the dynamically-driven variability as well as the essential driver of the variability. We does not conclude it but provide the suggestion of the possibility of the former case, considering the spatial location of OLR anomaly and low PV area in phase 6. This has been written in lines 324-330 in the original manuscript but the explanation may be not clear. We have revised lines 276-278 in Section 4 and a paragraph starting from line 336 in Section 5 in the revised manuscript.

2 Specific / minor comments

> title: Change to "geopotential height fields". In my opinion, the latter half of the title (".. using PV and geopotential...) could also be skipped, but this

**ACPD**
is a matter of taste, so I leave it to the authors to decide.

We apprieciate the suggestion. We have added 'height' to the original title.

> line 24: "to be dominant": consider rephrasing to "to be the dominant transport process"

We have revised line 24 as suggesteed.

> line 48-50: I wouldn't agree in that the paper by Nuetzel et al showed that the bimodality is a robust feature. Indeed, they showed that the bi-modality is very prominent only in older (NCEP) reanalysis data sets.

We appreciate the important comment.Our original manuscript does not correctly refer to their conclusion. We have revised lines 47-49 in the revised manuscript.

> line 113: This sentence makes it sound as if the low PV area in the anticyclone is usually conserved, but just not strictly, because of the "forcing processes such as deep convection". Diabatic heating and associated outflow from deep convection (divergent motion) is THE forcing process of the anticyclone, if it wasn't for that, there would be no low PV area to start off with. Therefore, I find this formulation a little weird. Please rephrase it to make the role of deep convection on forcing low PV more clear.

We meant to emphasize the effect of thickness variation as another factor which causes the change in the low PV area, besides of the diabatic heating. We have revised lines 110-113 of the revised manuscript to provide clearer explanation.
> line 130 / Equ. 2: On a similar note as major comment 1, a definition for the longitude-dependent quantity  $L(\lambda, t)$  along the lines suggested for  $\hat{A}$  could be given (i.e. as integral over  $\phi$ ).

We have added Eq. (4) for the definition.

> line 171: here, the authors state that the analysis of the timeseries in the preceding subsection "confirmed" that the dominant timescale of variability is the "quasi-biweekly" timescale. However, in the daily timeseries presented in Fig. 2, a monthly period is predominant, and the quasi-biweekly timescale is, if at all, only to be guessed "by eye". So either you have to weaken the statement here (e.g. indicates that quasi-biweekly variability can be identified from the timeseries) or make the analysis more quantitative (see also major comment 2).

We agree the statement 'The analysis regarding the total area so far confirmed  $\ldots$  ' was too strong. It was not confirmed in the previous section. We have revised the first paragraph of section 3.2.

> line 186: For 360 K, the flux is negative around its minimum, even within the range given by the standard deviation. So this argument holds only for Fig 3a (370 K), right?

We think it is still possible for the airmass to move both eastward and westward even the latitudinally averaged flux is mostly westward at 360K. But in that situation, the interpretation is more complex because the movement of passive tracer does not follow the movement of the airmass as there is a large nonzero PV source/sink. Therefore, we limit this statement only for 370K and 380K where the diabatic process is less significant. We have revised section 3.2 accordingly. **ACPD**
> line 193/194: "fixed latitude range in the southern part of the AMA" - in Garny et al, the total low PV area within 15-45 N was shown (see their Fig. 6), so this is not really a fixed latitude range, and neither only the southern part? The main difference is, apart from the slightly larger latitude range here, rather the weighted versus non-weighted low PV area, and the addition of the fluxes.

We agree that the statement was not accurate. The difference from Garny et al. is basically the thickness weighting that may give clearer picture to focus on both westward and eastward movement of the air. We have revised the description in line 205-207 of the revised manuscript.

> line 197: why only "as far as PV conservation holds"? Is the flux not valid if the the diabatic term is not equal zero? This would be worrisome for the whole analysis of the paper.

We meant that the interpretation using the simple relationship between the low PV air mass and its flux is valid under the assumption of the PV conservation. However, it is true that we can use the flux itself regardless of the convervation as you suggested. We have simply removed the phrase 'as far as PV conservation holds'.

> line 209: "dominant period of the variability in total low PV area" - actually show the power spectrum of the low PV area (see also comment on line 171, and major comment 2)?

The dominant period of the variability seen in the total low PV area has been found to be around 30 days in previous studies, and we do not mean to revisit it. We should have put the reference Garny and Randel (2013) here. We have added the reference, and revised the paragraph following your major comment.
> line 219: Do you mean to say that the timeseries filtered with a band pass filter within 5-20 days periods? Please specify.

Yes, that is what we did. We have added the description to the first paragraph of Section 4.1.

> line 232: "zonally averaged total perturbation variance": Do you mean the variance in terms of anomalies at each longitude, and then this variance is zonally averaged?

Yes we do. The original manuscript meant to describe the contribution of the leading two components to the total variance as a function of latitude. However, we reconsider that this statement is not necessary and decided to remove.

> line 233: the studies mentioned here rather analyzed tele-connections to the mid-latitudes than variability of the anticyclone itself, correct? So maybe it is not surprising that they find different pattern? (Also, correlation to the time-series "at a point at a midlatitude" is a bit vague - please clarify).

Yes. As they focus on midlatitude dynamical variability they used different methods and target variables and captured different variability patterns. In this part, it was found that the description "simple correlations with a time series at a point at a midlatitude" was not correctly described what has been done in Ding and Wang (2007). They made the composite analysis for the extreme events of positive geopotential height anomaly averaged over 35-45N, 55-75E. We are sorry for misleading. We have revised line 250-251 of the revised manuscript accordingly.

> I find the EOF analysis, and the PC lag analysis and life cycle a great approach to characterize the variability. Possibly, adding actual data points to Figure 9 to see the progression of the phases would be beneficial? Interactive comment

We appreciate the suggestion. The phase progress for the year 2016 has been added to Fig 9 as an example. Also, the caption of Fig. 9 has been revised.

> Fig. 11: agree that there is clear westward extension from phase 5 to 8, but does the low PV area "move back" to the east from phase 1 to 4, or is it shed? From the extend and strength of the low PV occurrence, it seems like the total PV area decrease over those phases. Related, is the total integral over the westward flux (in Fig. 12) equal to the sum of the eastward Flux over all phases? This would prove this point, and I guess it has to be the case, given that the total area appears to remain rather constant according to Fig. 13.

Our idea based on Figs. 12 and 13 is that, the low PV area mostly moves back to the east rather than 'shed' westward away from the main anticyclone vortex. Though this does not rule out the existence of some part of low PV actually shed and dissipated to the west, the shedding is likely to have only a small contribution to the budget of the air inside the anticyclone. This is supported by Figs. 12 and 13. The eastward and westward flux is nearly balanced on average over phases and the total PV area is nearly constant with the phase. The fact that the area of high low-PV probability decreases as phase progresses from 1 to 4 in Fig. 11 can be explained by the thickness variation. The average  $\sigma$  over 30N is larger than that over 20N on 370 K. Therefore, when the weighted low PV area does not change, the low PV airmass takes smaller area when it is moved northward in phase 2-4.

> Fig. 15: Not sure what the difference of black contours and color shading is - deviations from zonal mean versus anomalies from this deviation?

The black contours show the deviation from the longitudinal mean over the sector (0-150E) for each phase, which indicates the location of the anticyclone center. It shows
how the variability is large enough to change the anticyclone location The color shading shows the deviation from climatological mean, which indicates the longitudinal structure of the perturbation.

> lines 281-282: I'm not sure I understand the statement on the role of the subtropical jet on the deep gph anomalies. Please either remove, or add explanation/ citation for this statement.

What determines the vertical structure of the variability pattern is beyond the scope of this article. Therefore we have removed that sentence for simplicity.

> line 307: would you consider 30the "dominant variability"?

We consider the leading two EOF components corresponds to the most significant variability pattern in the area in a 5-20 day time scale. But the word 'dominant' may sound not suitable for the components which contribute to only about 30% of the total (normalized) variance. Therefore we have removed the word "dominant" from the sentense in line 336 of the revised manuscript.

> line 305/306: here you state that the west/east-ward flux of low PV is consistent with "eddy shedding", which could be true, but this notation implies that the eddies are at least partly "shed" from the Anticyclone, while your analysis seems to suggests that the total area is conserved during the quasi-biweekly variability, i.e. no actual "shedding" occurs. In Fig.3 b), where you show that the total eastward flux at 360 K is negative, does that imply actual westward shedding?

We used the word 'eddy shedding', in lines 252 and 305/306 in the original manuscript, in a broader context describing the westward movement and the detachment of the

**ACPD**
partial anticyclonic vortex from the main anticyclone, regardless of whether it is reversible or not. But later we stated in line 347 that they are mostly not actual 'shedding' at 370K level as you suggested, This might have made confusion, therefore we revised those lines without the notation 'eddy shedding'. About the low PV area budget at 360 K, there is still no clear answer from the result of this study. The source of the weighted low PV area on 360K is likely to be diabatic heating over the area of deep convection located overt the eastern part of the antiyclone. But its sink on the west of the anticyclone cannot be specified. The westward shedding may conrtibute, but it may be by diabatic heating/cooling . We have revised section 3.2 and include this discussion to line 217-225.

> general: State in all Figure caption, and possibly more often in the text, at which level the analysis is performed on! (I found this information to be rather hidden).

We are sorry for the insufficient information. We have added proper information of the isentropic and pressuer level in the text (in lines 189, 202, 273, and 274) and the figure captions (Figs. 4, 5, 6, 12, 13, and 14).

3 Typos / technical

- line 165: "pointed out by..." (add "out")
- line 220: dividing "by" their standard deviation (insert "by" )
- line 225: "longer months": change to "longer period" ?
- line 275: "rest phases": change to "for the rest of the phases" ?
- Fig. 15: Title and legend should say 15-25N and 35-45N rather then E.
- line 409: "noize": change to "noise"
- line 420: "persentage": change to "percentage"
- Fig. 4: which level?

**ACPD**
- Fig. 13: I would suggest to change/remove the heading"A west", as not only "A west" is shown

We have corrected the manuscript accordingly.

Derivation of Eqs. 2 and 3 (Eqs. 3 and 4 in the manuscript) :

Equation 2 can be derived in a way similar to the derivation of Eq. (13) of Butchart et al. (1986). For consistency, their notation is used in the following. They begin with the small change in the area  $\Delta A(t)_{\chi \ge \chi_0}$  enclosed by an isopleth  $\chi = \chi_0$ , denoted by  $\Gamma$ , with respect to the change in the contour position  $\Delta \vec{x}$ . Note that they assumed that the value of  $\chi$  increases inward of the area.

$$-\oint_{\Gamma} \Delta \vec{x} \cdot \frac{\nabla_{\theta} \chi}{|\nabla_{\theta} \chi|} ds = \Delta A(t)_{\chi \ge \chi_0} \tag{4}$$

To extend this formulation to the thickness-weighted area  $\hat{A}(t)_{\chi \ge \chi_0}$ , there needs an additional term for thickness change  $\Delta \sigma$  in the left hand side,

$$\int_{\chi \ge \chi_0} \Delta \sigma dA - \oint_{\Gamma} \Delta \vec{x} \cdot \frac{\nabla_{\theta} \chi}{|\nabla_{\theta} \chi|} \sigma ds = \Delta A(t)_{\chi \ge \chi_0}$$
(5)

Using the equation for thickness

$$\frac{\partial \sigma}{\partial t} + \nabla_{\theta} \cdot (\sigma \vec{v}) = -\frac{\partial}{\partial \theta} (\sigma \dot{\theta}), \tag{6}$$

the first term on the left hand side of Eq. (5) can be rewritten as follows.

$$\int_{\chi \ge \chi_0} \Delta \sigma dA = \Delta t \left[ -\int_{\chi \ge \chi_0} \nabla_{\theta} \cdot (\sigma \vec{v}) dA - \int_{\chi \ge \chi_0} \frac{\partial}{\partial \theta} (\sigma \dot{\theta}) dA \right] \qquad (7)$$

$$= \Delta t \left[ \oint_{\Gamma} \vec{v} \cdot \frac{\vec{\nabla}_{\theta} \chi}{|\vec{\nabla}_{\theta} \chi|} \sigma ds - \int_{\chi \ge \chi_0} \frac{\partial}{\partial \theta} (\sigma \dot{\theta}) dA \right] \qquad (8)$$

$$C15$$

**ACPD**
where  $\vec{v}$  is two-dimensional velocity, and  $\vec{\nabla}_{\theta}$  is two-dimensional gradient on an isentropic surface.

The second term on the right hand side is transformed using  $\Delta \vec{x} \cdot \vec{\nabla}_{\theta} \chi \simeq -\Delta t \cdot \partial \chi / \partial t$  as shown in ?. Then by applying  $\Delta t \rightarrow dt$  we obtain the following,

$$\frac{d}{dt}\hat{A}(t)_{\chi\geq\chi_0} = \oint_{\Gamma} \left(\frac{\partial\chi}{\partial t} + \vec{v}\cdot\nabla_{\theta}\chi\right) \frac{\sigma ds}{|\nabla_{\theta}\chi|} - \int_{\chi\geq\chi_0} \frac{\partial}{\partial\theta}(\sigma\dot{\theta})dA \tag{9}$$

As the first integral contains the advection term, the bracket can be replaced with a nonconservation term F.

$$\frac{\partial \chi}{\partial t} + \vec{v} \cdot \nabla_{\theta} \chi = F \tag{10}$$

When  $\chi$  is potential vorticity and the area A is defined to have potential vorticity below the reference value, Eq. (2) is obtained. The subgrid scale mixing term in Butchart et al. (1986) comes from the difference between the true divergence term and the divergence term calculated from resolved variables. Although our equation does not have divergence term, we consider it is still better to include unresolved effect such as subgrid scale mixing, which is included in F. Then, when  $\chi$  is potential vorticity and the area A is defined to have potential vorticity below the reference value, using

$$F = -q\frac{\partial\dot{\theta}}{\partial\theta} + \dot{\theta}\frac{\partial q}{\partial\theta} + (\text{unresolved term})$$
(11)

thus we obtain Eq. (2).

Equation (2) can also be derived from the general mass conservation expression in a PV- $\theta$  coordinate introduced in Nakamura (1995; https://doi.org/10.1175/1520-
$$\left(\frac{\partial m}{\partial t}\right)_{q,\theta} + \left(\frac{\partial \mathcal{M}(\dot{q})}{\partial q}\right)_{\theta,t} + \left(\frac{\partial \mathcal{M}(\dot{\theta})}{\partial \theta}\right)_{q,t} = 0$$
(12)

where  $m = \mathcal{M}(1)$  and thickness-weighted area integration operator  $\mathcal{M}$  is defined as

$$\mathcal{M}(*) = \int \int_{q \le q_0} (*)\sigma dA = \int_{q^* \le q} dq^* \oint_{q^*} (*) \frac{\sigma ds}{|\nabla_\theta q^*|}$$
(13)

Substituting  $\dot{q} = F$  and expand  $\mathcal{M}(\dot{q})$  and  $\mathcal{M}(\dot{\theta})$ , we obtain Eq. (2).

Next, let us derive Eq. (3). Now suppose the small change of the area  $\Delta \hat{A}(t)_{\chi \geq \chi_0, \lambda \leq \lambda_0}$ enclosed by the isopleth  $\chi = \chi_0$  and the circle of longitude  $\lambda_0$ . Let  $\Gamma_q$  and  $\Gamma_l$  respectively be the isopleth and the circle of longitude consisting the border of the area as shown in Fig. 1. Equation (4) is modified as follows,

The integral of the first term on the left hand side is performed over the area  $\Delta \hat{A}(t)_{\chi \geq \chi_0, \lambda \leq \lambda_0}$ , and the line integral of the second term is performed only over  $\Gamma_q$ , as  $\Gamma_l$  is constant with time. The first term can be rewritten as follows, using the Gauss'theorem for  $\Gamma_q$  and  $\Gamma_l$ ,

$$\begin{split} \int_{\chi \ge \chi_0, \lambda \le \lambda_0} \Delta \sigma dA &= \Delta t \left[ -\int_{\chi \ge \chi_0} \nabla_{\theta} \cdot (\sigma \vec{v}) dA - \int_{\chi \ge \chi_0} \frac{\partial}{\partial \theta} (\sigma \dot{\theta}) dA \right] & (14) \\ &= \Delta t \left[ \int_{\Gamma_q} \vec{v} \cdot \frac{\vec{\nabla}_{\theta} \chi}{|\vec{\nabla}_{\theta} \chi|} \sigma ds + \int_{\Gamma_l} u \sigma ds - \int_{\chi \ge \chi_0} \frac{\partial}{\partial \theta} (\sigma \dot{\theta}) dA \right] & (15) \end{split}$$

Then we obtain Eq. (3) with the additional term  $\hat{F}(\lambda)$  defined as follows. The integral
is performed over the longitude circle  $\lambda_0$  consisting the border of the area.

$$\hat{F}(\lambda) = \int_{q \le q_0, \lambda_0} u\sigma ds = \int_{q \le q_0, \lambda_0} u\sigma R d\phi$$
(16)

---

## Author Comment (AC2) · 24 Aug 2020

We greatly appreciate the reviewer's invaluable and constructive comments. We have revised our manuscript following your and the other reviewer's comments. Responses to each of the major and minor comments are written below.

Major comments

> It is unclear whether it is a major or minor point but the basic equation

(1) which is taken from Garny and Randel (2013) is technically wrong as it is presented. The integrand dS in the first member of the r.h.s. is not a line element but a line element divided by the modulus of the horizontal gradient of the PV. This is stated by Garny and Randel and, otherwise, the equation is not even dimensionally homogeneous. I hope that this detail has not been missed by the authors and that the error was only introduced during writing but it is quite worrying.

It is true that the factor of a line element was missing in Eqs. 1, 3, and 4. Additionally, we also found another missing term which should be added in the right hand side of Eqs. 3 and 4. Correct forms of those equations are as follows. Their derivations are added to Appendix B of the revised manuscript and also at the bottom of this reply. We are sorry for the incorrect description. This correction does not affect our analysis results in this study, as the integral terms are not directly used.

$$
\frac{d}{dt}\hat{A}_{\text{tot}}(t) = \oint_{q=q_0}\left(-q\frac{\partial\dot{\theta}}{\partial\theta} + \dot{\theta}\frac{\partial q}{\partial\theta}\right)\frac{\sigma dS}{|\nabla_\theta q|} - \int_{q\le q_0}\frac{\partial}{\partial\theta}(\sigma\dot{\theta})dA ,
$$
$$
+ \ (\text{unresolved term}) \tag{1}
$$

$$
\frac{d}{dt}\hat{A}_{\text{west}}(t) = -\hat{F}(\lambda_0) + \oint_{q=q_0,\lambda\le\lambda_0}\left(-q\frac{\partial\dot{\theta}}{\partial\theta} + \dot{\theta}\frac{\partial q}{\partial\theta}\right)\sigma dS - \int_{q\le q_0,\lambda\le\lambda_0}\frac{\partial}{\partial\theta}(\sigma\dot{\theta}),
$$
$$
+ \ (\text{unresolved term}) \tag{2}
$$

> The main claim of the manuscript is that the oscillations are mostly of dynamical origin and reversible, and that forcing by convection and dissipation are not involved. This is quite opposite to conclusions of previous works

and also to Wei et al. (2019, doi:10.1029/2019GL086180) and references herein which is another stream of research that should be quoted and discussed. Another relevant work that studies PV fluxes on isentropic surfaces is Ortega et al. (2018, doi: 10.1002/qj.3261) which is also missing in the reference and should be used to compare the results of the manuscript.

About the causal relationship between convective forcing (particularly over Indian sector) and the dynamical variability in the upper troposphere, we do not very much agree with the argument of Wei et al. for two reasons. First, their result that the negative rainfall anomaly over northern India precedes the key day of the anticyclone east-west oscillation may be a consequence of the choice of the key day and the definition of the variability index, a difference between area-averaged GPH over Tibetan Plateau and Iranian Plateau. Second, they did not provide convincing mechanism of convection anomaly over northern India driving the dynamical field anomalies including midlatitudes. They referred to Karmakar et al. (2017), and Ding and Wang (2007), Karmakar et al. did not propose physical process responsible for it. Ding and Wang suggested a mutual positive feedback, in which midlatitude upper level circulation anomalies enhances the convection and in turn the convection triggers the midlatitude Rossby wave train. Thus they do not exclude the possibility that the convection anomaly is forced by the upper level circulation anomalies. The results of a composite analysis in Wei et al. (their Fig. 3) and ours (Fig. 14) do not necessarily agree for the feature of convection anomalies. The difference may come from the different reference index and the different pressure level. The discussion about this point has been added to the revised manuscript. We have added the reference to their work in line 350 in discussion section.

The formulation used in Ortega et al. has some similarity to ours, but it used the different definition of flux for the different purpose. They focus on PV value and its thickness-weighted flux in latitudinal direction, whereas our study focuses on the weighted area enclosed by a PV contour, and longitudinal flux of that area (mass). Their definition

cannot well quantify the movement of the air inside the anticyclone as the PV-based definition can do. We have added the reference to Ortega et al. (2018) and this explanation in line 156-158 of the revised manuscript.

> The manuscript focuses on the zonal mass flux of air with low PV and finds that the mean flux at 370K oscillates around zero over the range of latitudes of the AMA. This is basically the main result. However, this does not mean that there is no zonal flux of PV. It is clear from fig.11 that during the phase of eastward flux at 60E, the air carries less negative PV that during the phase of westward flux. Therefore the zero mean mass flux does not rule out a non zero mean PV flux, where negative PV is created on the east by convection and dispersed and lost to the background on the west by vortex shedding within a biweekly cycle. It is useful to notice that the circulation time around the anticyclone and its erosion rate are also of about two weeks (Legras and Bucci, 2019, doi: 10.5194/acp-2019-1075). PV is clearly not well conserved during the bi-weekly cycle.

We appreciate reminding the important difference between the conservation of the low PV area and the conservation of PV values. The description in the original manuscript which is based on PV conservation is not appropriate in this sense. Our discussion is based on the budget of the weighted low PV area. This view has better implication for the variability of chemical tracers, given that the threshold PV value properly reflects the mixing barrier. We have revised lines 158-160 in the revised manuscript to make this point clear.

Minor comments

> l. 51-53: Is it so clear that the two questions are well separated?

These two questions can be treated separately, because the intensity and position/structure of the AMA can be quantified independently with each other.

> l. 63 "is often"

We have revised the sentence in this line as suggested.

> l. 141: I do not see why the divergence term disappears in this equation.

We have added the derivation of Eqs. 3 and 4 in Appendix B in the revised manuscript.

> l. 145: The integral is at fixed longitude and the integrand is latitude over the range where the PV is below the threshold and F is the mass flux (rather than the movement) of low PV air across a given latitude. This is badly described and the scheme in fig.1c adds to the confusion.

We have revised lines 148-149 in the revised manuscript as suggested.

> l. 164: I do not see the need for a 31-day filter when the average is done over 38 years. This should be enough to scramble the phases of the AMA oscillations.

We used a low-pass filter to ensure that it represents the mean seasonal variation, although it does not change much of the result.

> l. 173: At this stage, the evidence is only based on the visual appearance of a single year record.

The statement in line 173 of the original manuscript 'The analysis regarding the total area so far confirmed . . . ' was not appropriate. We have revised the first paragraph of section 3.2.

> l. 176: This line should refer to eq. (5) if this is what is shown.

We have added the reference of the equation.

> l. 181: This line makes me worrying whether the total heating, including latent heating, is accounted as it should in this work or whether only radiative heating is used. At 370K, it is however correct to assume that radiative heating dominates.

The notation 'differential radiative heating' was misleading, as other processes such as latent heating should be accounted as well. We have revised the line 193 of the revised manuscript.

> l. 190: I asume that the results are shown on the 370 K surface but this should be stated. It is very difficult to distinguish the blue and red contours in fig. 4. The text mention that fig.4 shows the mass weighted length that should have dimension kg/(K x m) and the caption says that it is a weighted area with dimension kg/K. Please clarify. Provide a definition for this weighted area that depends on the longitude and discard L if is not used.

The value shown in Figure 4 has a unit of kg/K and factor of $10^{1}2$. It is the weighted area calculated over each grid area and summed up for latitudinal direction. We have revised the figure caption and the description in lines 201-202. We also have made blue contours in Fig. 4 thicker and dashed to make them more visible.

> In the sequel, no PV diagnostic is shown on other surfaces than 370K. The choice of 370K is justified in the appendix on the basis of the best

definition of AMA in terms of PV but is would nevertheless be interesting to look at over surfaces. 380 K was privileged in Ploeger et al. (2015) and 360 K is closer to the level where convective detrainement is the strongest. It is also where the mean eastward and westward branches of the AMA are maximum and where the isentropic divergence is maximum.

We have revised Figs. 4 and 5 to show the results at three different levels 360, 370, and 380 K. As expected, the contribution of the longitudinal advection to the low PV area distrubution is less clear at 360 K at 380K, as there are more significant source and sinks. The whole picture of the budget of low PV airmass is more complicated than what is seen at a single level, we consider the longitudinal ocsillatory behavior with a time scale of quasi-biweekly is one important feature, and the low PV air at 370 K is the best proxy for that feature. We have added these discussions to section 3.2 of the revised manuscript.

> l. 197: I do not see why PV conservation is invoked here. It is clear that PV is not well conserved here (see fig.11).

As the other reviewer also pointed out, the PV conservation needs not be assumed here for the use of the longitudinal flux of the low PV area. We have removed that phrase from the original manuscript.

> l. 209 and Fig. 6: How should we interpret the significance curves on Fig. 6? The peak is not that strong and shows there is a plateau in the spectrum intensity between 9 days and 25 days. 9 days is more a cutoff period than a dominating period.

We agree that the peak in Fig. 6 does not show clear characteristic time scale of 'quasi-biweekly'. There is broad range of possible characteristic time scale centered around quasi-biweekly. We have revised lines 226-228 in the revised manuscript.

> Sect. 4 It is a bit surprising that the study switches here to the geopotential on the 100 hPa surface. Having done all the work to interpolate basic variables on isentropic surfaces would have made easy to calculate the Montgomery potential on such surfaces. Basically, the results would not have been very different but this would have been more consistent, especially because isentropic and isobaric surfaces may differ quite significantly in the Asian monsoon region. As the authors are looking for a cycle, they should have considered the MSSA method which is particularly well suited (Ghil et al, 2012, doi:10.1029/2001RG000092) and would have saved time and space.

The use of Montgomery potential may be consistent in our analysis. However, we rather chose geopotential at the 100 hPa pressure surface for the EOF analysis in order to make the comparison with earlier studies easier, as most studies on the bimodality of the AMA center location used 100 hPa geopotential height. The result would be similar if we perform the analysis using Montgomery streamfunction on 380 or 390 K level. The extended EOF analysis is mathematically equivalent to MSSA (Plaut and Vautard, 1994) and more often used in geophysical studies in which the number of spatial grid points are much larger than the number of time steps in a cycle. For example, Wang and Duan (2015) used extended EOF for 10-20 day filtered diabatic heating in the Asian monsoon region. We did extended EOF and complex EOF analyses and got the similar results to the case of EOF analysis, as mentioned in line 225 in the original manuscript. We only show the result of EOF analysis for simplicity.

> l. 220 "dividing by their"

We have revised the sentence in line 236 in the revised manuscript as suggested.

> l.220 Why the square root of grid area and not the area in the weight?

When the data is weighted by the area, the variance or correlation should be multiplied by the area. This corresponds to multiplying the anomaly by the square root of the area.

> l. 240: It is quite difficult to understand fig. 9 which is introduced in a section where PV plays no role. Please improve the caption such that it makes sense when the reader is at line 240 in the text.

The caption of Fig.9 was incorrectly input. We are very sorry for the mistake. We have replaced it with the correct caption.

> Sect. 4.2 The first paragraph concludes that variability is determined by internal inviscid and adiabatic dynamics but the second paragraph shows there is a pattern of convection associated with the oscillation which some- what contradicts the first paragraph if we admit that convection does not only react passively but generates a forcing. The authors do not attempt to provide a balanced view and just discard the convective influence in this section and in the conclusions l.323-330.

About the role of the convection variability, we does not conclude but suggest the pos- sibility that the dynamical variability drives the convection variabilty, considering the spatial location of OLR anomaly and low PV area in phase 6. This has been written in lines 324-330 in the original manuscript but the explanation may not be clear. We have revised lines 276-278 in Section 4 and a paragraph starting from line 336 in Section 5 in the revised manuscript.

Derivation of Eqs. 1 and 2 (Eqs. 3 and 4 in the manuscript) :

Equation (1) can be derived in a way similar to the derivation of Eq. (13) of Butchart et al. (1986). For consistency, their notation is used in the following. They begin with the small change in the area $\Delta A(t)_{\chi \geq \chi_0}$ enclosed by an isopleth $\chi = \chi_0$, denoted by $\Gamma$, with respect to the change in the contour position $\Delta \vec{x}$. Note that they assumed that the value of $\chi$ increases inward of the area.

$$-\oint_{\Gamma} \Delta \vec{x} \cdot \frac{\nabla_{\theta}\chi}{|\nabla_{\theta}\chi|} ds = \Delta A(t)_{\chi \geq \chi_0} \qquad (3)$$

To extend this formulation to the thickness-weighted area $\hat{A}(t)_{\chi \geq \chi_0}$, there needs an additional term for thickness change $\Delta \sigma$ in the left hand side,

$$\int_{\chi \geq \chi_0} \Delta \sigma dA - \oint_{\Gamma} \Delta \vec{x} \cdot \frac{\nabla_{\theta}\chi}{|\nabla_{\theta}\chi|} \sigma ds = \Delta A(t)_{\chi \geq \chi_0} \qquad (4)$$

Using the equation for thickness

$$\frac{\partial \sigma}{\partial t} + \nabla_{\theta} \cdot (\sigma \vec{v}) = -\frac{\partial}{\partial \theta}(\sigma \dot{\theta}), \qquad (5)$$

the first term on the left hand side of Eq. (4) can be rewritten as follows.

$$\int_{\chi \geq \chi_0} \Delta \sigma dA = \Delta t \left[ -\int_{\chi \geq \chi_0} \nabla_{\theta} \cdot (\sigma \vec{v}) dA - \int_{\chi \geq \chi_0} \frac{\partial}{\partial \theta}(\sigma \dot{\theta}) dA \right] \qquad (6)$$

$$= \Delta t \left[ \oint_{\Gamma} \vec{v} \cdot \frac{\vec{\nabla}_{\theta}\chi}{|\vec{\nabla}_{\theta}\chi|} \sigma ds - \int_{\chi \geq \chi_0} \frac{\partial}{\partial \theta}(\sigma \dot{\theta}) dA \right] \qquad (7)$$

where $\vec{v}$ is two-dimensional velocity, and $\vec{\nabla}_{\theta}$ is two-dimensional gradient on an isentropic surface.

The second term on the right hand side is transformed using $\Delta\vec{x} \cdot \vec{\nabla}_\theta\chi \simeq -\Delta t \cdot \partial\chi/\partial t$ as shown in ?. Then by applying $\Delta t \rightarrow dt$ we obtain the following,

$$\frac{d}{dt}\hat{A}(t)_{\chi \geq \chi_0} = \oint_\Gamma \left(\frac{\partial\chi}{\partial t} + \vec{v}\cdot\nabla_\theta\chi\right)\frac{\sigma ds}{|\nabla_\theta\chi|} - \int_{\chi \geq \chi_0}\frac{\partial}{\partial\theta}(\sigma\dot{\theta})dA \tag{8}$$

As the first integral contains the advection term, the bracket can be replaced with a nonconservation term $F$.

$$\frac{\partial\chi}{\partial t} + \vec{v}\cdot\nabla_\theta\chi = F \tag{9}$$

When $\chi$ is potential vorticity and the area $A$ is defined to have potential vorticity below the reference value, Eq. (1) is obtained. The subgrid scale mixing term in Butchart et al. (1986) comes from the difference between the true divergence term and the divergence term calculated from resolved variables. Although our equation does not have divergence term, we consider it is still better to include unresolved effect such as subgrid scale mixing, which is included in $F$. Then, when $\chi$ is potential vorticity and the area $A$ is defined to have potential vorticity below the reference value, using

$$F = -q\frac{\partial\dot{\theta}}{\partial\theta} + \dot{\theta}\frac{\partial q}{\partial\theta} + (\text{unresolved term}) \tag{10}$$

thus we obtain Eq. (1).

Equation (1) can also be derived from the general mass conservation expression in a PV-$\theta$ coordinate introduced in Nakamura (1995) ;

$$\left(\frac{\partial m}{\partial t}\right)_{q,\theta} + \left(\frac{\partial\mathcal{M}(\dot{q})}{\partial q}\right)_{\theta,t} + \left(\frac{\partial\mathcal{M}(\dot{\theta})}{\partial\theta}\right)_{q,t} = 0 \tag{11}$$

where $m = \mathcal{M}(1)$ and thickness-weighted area integration operator $\mathcal{M}$ is defined as

$$\mathcal{M}(*) = \int\int_{q \leq q_0} (*)\sigma dA = \int_{q^* \leq q} dq^* \oint_{q*} (*)\frac{\sigma ds}{|\nabla_\theta q^*|} \tag{12}$$

Substituting $\dot{q} = F$ and expand $\mathcal{M}(\dot{q})$ and $\mathcal{M}(\dot{\theta})$, we obtain Eq. (1).

Next, let us derive Eq. (2). Now suppose the small change of the area $\Delta \hat{A}(t)_{\chi \geq \chi_0, \lambda \leq \lambda_0}$ enclosed by the isopleth $\chi = \chi_0$ and the circle of longitude $\lambda_0$. Let $\Gamma_q$ and $\Gamma_l$ respectively be the isopleth and the circle of longitude consisting the border of the area as shown in Fig. 1. Equation (3) is modified as follows,

The integral of the first term on the left hand side is performed over the area $\Delta \hat{A}(t)_{\chi \geq \chi_0, \lambda \leq \lambda_0}$, and the line integral of the second term is performed only over $\Gamma_q$, as $\Gamma_l$ is constant with time. The first term can be rewritten as follows, using the Gauss'theorem for $\Gamma_q$ and $\Gamma_l$,

$$\int_{\chi \geq \chi_0, \lambda \leq \lambda_0} \Delta \sigma dA = \Delta t \left[ -\int_{\chi \geq \chi_0} \nabla_\theta \cdot (\sigma \vec{v}) dA - \int_{\chi \geq \chi_0} \frac{\partial}{\partial \theta}(\sigma \dot{\theta}) dA \right] \tag{13}$$

$$= \Delta t \left[ \int_{\Gamma_q} \vec{v} \cdot \frac{\vec{\nabla}_\theta \chi}{|\vec{\nabla}_\theta \chi|} \sigma ds + \int_{\Gamma_l} u\sigma ds - \int_{\chi \geq \chi_0} \frac{\partial}{\partial \theta}(\sigma \dot{\theta}) dA \right] \tag{14}$$

Then we obtain Eq. (2) with the additional term $\hat{F}(\lambda)$ defined as follows. The integral is performed over the longitude circle $\lambda_0$ consisting the border of the area.

$$\hat{F}(\lambda) = \int_{q \leq q_0, \lambda_0} u\sigma ds = \int_{q \leq q_0, \lambda_0} u\sigma R d\phi \tag{15}$$

Reference: Nakamura, N. (1995). Modified Lagrangian-mean diagnostics of the stratospheric polar vortices. Part I. Formulation and analysis of GFDL SKYHI GCM. Journal of the Atmospheric Sciences, 52(11), 2096–2108.

[Figure]

[Figure]

**Fig. 1.**